# Charge and Antipodal Matching across Spatial Infinity

Federico Capone$^{\mathscr{I}^-}$, Kevin Nguyen$^{i^0}$ and Enrico Parisini$^{\mathscr{I}^+}$

$^{\mathscr{I}^\pm}$*Mathematical Sciences and STAG Research Centre, University of Southampton,*
*Highfield, Southampton SO17 1BJ, UK*
$^{i^0}$*Department of Mathematics, King's College London,*
*The Strand, London WC2R 2LS, UK*

federico.capone@soton.ac.uk, kevin.nguyen@kcl.ac.uk, e.parisini@soton.ac.uk

## Abstract

We derive the antipodal matching relations used to demonstrate the equivalence between soft graviton theorems and BMS charge conservation across spatial infinity. To this end we provide a precise map between Bondi data at null infinity $\mathscr{I}$ and Beig–Schmidt data at spatial infinity $i^0$ in a context appropriate to the gravitational scattering problem and celestial holography. In addition, we explicitly match the various proposals of BMS charges at $\mathscr{I}$ found in the literature with the conserved charges at $i^0$.

# 1   Introduction

> *In an $\mathcal{S}$-matrix approach where data on $\mathscr{I}^+$ are related to the*
> *ones on $\mathscr{I}^-$, the point $i^0$ has presumably to play a role.*
> R. Beig, 1984.

Important insights into the theory of gravitational scattering in asymptotically flat space-times have been recently accumulated as a result of Strominger's key observation that con-servation of the charges associated with the Bondi, van der Burg, Metzner and Sachs (BMS) asymptotic symmetries [1–3] implies Weinberg's leading soft graviton theorem [4,5]. Follow-ing this finding, new asymptotic symmetries and soft theorems have been discovered [6–11] and a new approach to flat space scattering amplitudes referred to as *celestial holography* has been put forward [12–16].

These developments offer new promising prospects in the study of (quantum) gravity beyond the perturbative regime. In order for the program of celestial holography to reach its full potential, we should carefully set it up in a way which appropriately incorporate the

nonlinear nature of General Relativity. Indeed generic asymptotically flat spacetimes significantly differ from Minkowski space, especially with regards to their structure at spatial infinity $i^0$. Physical fields are generically not single-valued at $i^0$, such that continuity cannot be invoked in order to relate their behavior from past null infinity $\mathscr{I}^-$ to future null infinity $\mathscr{I}^+$. These considerations should play an important role in celestial holograhy. Indeed the newly discovered connections between asymptotic symmetries in General Relativity and soft graviton theorems crucially rely on $i$) the definition of a single BMS group acting simultaneously on both $\mathscr{I}^+$ and $\mathscr{I}^-$ via antipodal identifications of the symmetry generators and asymptotic fields, and on $ii$) the conservation of BMS charges from $\mathscr{I}^-_+$ to $\mathscr{I}^+_-$ across $i^0$. Validity of these two conditions in the nonlinear theory should be tightly connected with the behavior of the gravitational field in a neighborhood of $i^0$.

In the present work we wish to shed further light on the matching of BMS charges across spatial infinity $i^0$ and on the corresponding antipodal matching conditions *in the context relevant to the gravitational scattering problem*. The class of spacetimes typically considered in that context are a peeling version of those studied by Christodoulou and Klainerman (CK), which constitute a set of asymptotically flat geometries non-linearly close to Minkowski space[1] [17]. Although this class of spacetimes satisfies conditions i) and ii) as defined above, they certainly do not contain all configurations of interest. In particular, they do not account for spacetimes with nonzero supertranslation charges at $i^0$ [18]. Thus a nontrivial matching of the charges requires one to consider a broader class of spacetime asymptotics. We will however not investigate how and whether these asymptotics result from the evolution of mathematically well-defined initial data sets. See the work of Mohamed and Valiente Kroon along these lines in the case of spin-1 and spin-2 fields [19].

A key result of our approach is the mapping of scattering data at $\mathscr{I}$ to gravitational data in a neighborhood of $i^0$. Our treatment is entirely coordinate-based and relies on the Bondi–Sachs description of the gravitational field near $\mathscr{I}$ [1,2,20] and on the Beig–Schmidt description of the gravitational field near $i^0$ [21,22]. It therefore differs from the recent work of Prabhu and Shehzad [23,24] who studied the matching of charges within the Ashtekar–Hansen formalism set up to treat $i^0$ and $\mathscr{I}$ simultaneously [25,26]. Rather we proceed by performing an asymptotic coordinate transformation between Bondi and Beig–Schmidt gauges in order to obtain an explicit map relating the respective asymptotic data.

The descriptions of the gravitational field at $\mathscr{I}$ or $i^0$ have distinctive features, and relating them is therefore of high interest. On the one hand, spatial infinity is the locus where the

---

[1]The initial data sets considered by CK are characterised by a spherically symmetric mass parameter and result in spacetimes which do not satisfy the peeling property.

variational principle is well-defined, and charges are both integrable and conserved. The most general phase space in Beig-Schmidt gauge was analysed by Compère and Dehouck (CD) [27] (see also [28–30]). Their charges satisfy all the desirable properties[2] and give a faithful representation of the BMS algebra without central extension. Their treatment also extends previous constructions [25, 31, 32, 34–36] in that they account for nonzero leading electric and magnetic Weyl tensor $E_{ab}$ and $B_{ab}$, and do not impose parity conditions on the corresponding potentials $\sigma$ and $k_{ab}$. On the other hand, charges computed at $\mathscr{I}$ are neither integrable nor conserved, a fact closely related to the leakage of symplectic flux through $\mathscr{I}$ in the form of gravitational radiation.

Inspired by the celestial holography literature, we assume the scattering data at $\mathscr{I}$ to admit a polynomial expansion in negative powers of the radial and retarded time coordinates $r$ and $u$, and no radiation in the limit to $i^0$. *This is the notion of gravitational scattering we consider in this work.* Under these assumptions we find that the scattering data maps onto a restricted subset of the CD phase space. In particular both $\sigma$ and $k_{ab}$ turn out to satisfy specific parity conditions, although the well-posedness of Einstein's equations near spatial infinity does not require them a priori. We also demonstrate that the resulting magnetic Weyl tensor $B_{ab}$ vanishes, a property previously assumed by Prabhu and Shehzad when considering the matching of Lorentz charges [24]. As far as we know, $B_{ab} = 0$ had only been established for spacetimes that are axisymmetric or stationary [37]. We confirm that it is in fact a feature of the gravitational scattering problem (as considered here).

The explicit map between Bondi and Beig–Schmidt data allows us to derive the antipodal matching relations together with the conservation of the BMS charges across spatial infinity. This in particular *implies* that only the diagonal subgroup of BMS($\mathscr{I}^+$)$\times$ BMS($\mathscr{I}^-$) is a symmetry of the entire spacetime asymptotic structure, as it was originally assumed by Strominger in his seminal work [4].

Various analyses of the requirements under which $\mathscr{I}^+$ and $\mathscr{I}^-$ and their symmetries can be matched across spatial infinity can be found in the literature. Perhaps the first step in this

---

[2]Note that the renormalization procedure that CD propose involves a Mann-Marolf-type counterterm [31, 32]. This prescription is well-known to partially break bulk diffeomorphisms, in the sense that in $d = 4$ it requires additional boundary conditions on the gravitational fields depending on the choice of the regulating surfaces in order for the variational principle to be well-defined. With our boundary conditions, this issue does not play any role. It is however not clear whether in principle it is possible to define alternative schemes that fully preserve covariance at spatial infinity. Another long-known problematic feature of renormalization at spatial infinity is the non-locality of counterterms, as first stressed in [33]. From a holographic point of view, this may be due to either some form of incompleteness of the current perspectives or a fundamental property of theories dual to flat spaces. It would be thus interesting to assess more in depth these two points.

direction was taken by Herberthson and Ludvigsen in demonstrating the antipodal matching of the Bondi mass aspect [38]. More recently Troessaert derived Strominger's original antipodal matching condition relating the supertranslation symmetry parameters of BMS($\mathscr{I}^+$) and BMS($\mathscr{I}^-$) [4,39]. In subsequent work Henneaux and Troessaert studied a set of parity conditions in the Hamiltonian formulation of gravity that allows for a canonical realisation of BMS symmetries at $i^0$ and argued that such a phase space supports Strominger's antipodal matching condition [40–42]. The aforementioned analysis of Prabhu and Shezad is instead framed within the Ashtekar–Hansen formalism and focuses on the matching of the charges themselves [23, 24]. The present paper builds upon this literature by giving the complete map of asymptotic data and charges between $\mathscr{I}^+_-$ and $\mathscr{I}^-_+$.

From our analysis it also follows that the various proposals of BMS charges in Bondi gauge found in the literature [43–48] all match with the conserved charges at spatial infinity. This is a consequence of the fact that the terms by which they differ vanish in the limit to $i^0$ under our working assumptions.

In addition, our work highlights the restrictions on the global spacetime asymptotic structure resulting from a choice of data at $\mathscr{I}$. This turns into a signpost indicating the limits of validity of the standard scattering setup, as well as a pathway to envision extensions towards a more general holographic framework. Indeed we have not restricted our analysis to solutions that are close to Minkowski space in the sense of CK. For these reasons, we believe that our approach is naturally suited to study the interplay between null and spatial infinity and to explore phenomenologically relevant processes beyond perturbative quantum gravity.

The paper is organized as follows. In section 2 we recall the features of asymptotically flat gravity at null infinity $\mathscr{I}$ in Bondi gauge and introduce assumptions regarding the behavior of the fields in their limit to $\mathscr{I}^-_+$ and $\mathscr{I}^+_-$. In section 3 we describe the Beig–Schmidt framework used to deal with Einstein's equations in a neighborhood of spatial infinity $i^0$. In section 4 we present the map from the Bondi gauge to the Beig–Schmidt gauge, whose details are given in appendix B. In section 5 we use this map together with Einstein's equations to derive the antipodal matching relations of the Bondi mass aspect, angular momentum aspect and shear tensor. In section 6 we provide the matching of BMS charges between null and spatial infinity.

**Conventions.** Three-dimensional indices are denoted with Roman letters $a, b, c, ..., h_{ab}$ is the metric of the three-dimensional de Sitter spacetime $\mathcal{H}$ and $D_a$ is its compatible covariant derivative. The metric on the celestial sphere $\mathbb{S}^2$ is denoted with $\gamma_{AB}$ and $\nabla_A$ is the covariant

derivative, capital Roman indices label the coordinates on this manifold. Covering the sphere $\mathbb{S}^2$ with angular coordinates $x^A = (\theta, \varphi)$, we define the *antipodal map* $\Upsilon(\theta, \varphi) = (\pi - \theta, \varphi + \pi)$. A tensor $T$ on $\mathbb{S}^2$ is of *odd parity* under $\Upsilon$ if $\Upsilon^* T = -T$. In terms of components of a vector field for example, this means that $T_\varphi$ and $T_\theta$ are odd and even functions on the sphere, respectively. Even parity under $\Upsilon$ is similarly defined. Covering $\mathcal{H}$ with global coordinates $x^a = (\tau, \theta, \varphi)$, we also define the *$\mathcal{H}$-antipodal map* $\Upsilon_{\mathcal{H}}(\tau, \theta, \varphi) = (-\tau, \pi - \theta, \varphi + \pi)$ and analogous considerations on parity properties apply.

## 2 Gravity at null infinity

Our main objective is to derive relations between quantities defined at past null infinity $\mathscr{I}^-$ and at future null infinity $\mathscr{I}^+$, more specifically in their limit to spatial infinity $i^0$. We will describe the gravitational field at null infinity in Bondi gauge, where the metric takes the form

$$ds^2 = g_{uu}\, du^2 + 2\, g_{ur}\, du\, dr + 2\, g_{uA}\, du\, dx^A + g_{AB}\, dx^A\, dx^B \,, \tag{2.1}$$

where $r$ is a luminosity distance, $u$ is a retarded time and $x^A$ are coordinates covering the celestial sphere $\mathbb{S}^2$. The limits $r \to \infty$ and $r \to \infty, u \to -\infty$ describe the approach to $\mathscr{I}^+_+$ and $\mathscr{I}^+_-$, respectively. A similar gauge can be adopted near $\mathscr{I}^-$ in terms of an advanced time coordinate $v$, where $v \to \infty$ describes the approach to $\mathscr{I}^-_+$. Following the conventions of Strominger [4], the Bondi gauge at $\mathscr{I}^-$ is formally obtained by applying the transformation $u \mapsto -v$ to (2.1) and all subsequent equations. Assuming that the large-$r$ expansion does not contain logarithmic terms, Einstein's equations are solved by

$$g_{uu} = -1 + \frac{2m}{r} + \frac{\phi}{r^2} + O(r^{-3})\,, \tag{2.2a}$$

$$g_{ur} = -1 + \frac{1}{16r^2} C_{AB} C^{AB} + O(r^{-3})\,, \tag{2.2b}$$

$$g_{uA} = \frac{1}{2}\nabla^B C_{AB} + \frac{2}{3r}\left(N_A + u\,\partial_A m - \frac{3}{32}\partial_A\left(C_{BC}C^{BC}\right)\right) + O(r^{-2})\,, \tag{2.2c}$$

$$g_{AB} = r^2\,\gamma_{AB} + r\,C_{AB} + \frac{1}{4}\gamma_{AB}\,C_{CD}C^{CD} + O(r^{-1})\,. \tag{2.2d}$$

The quantities $m, N_A$ and $C_{AB}$ are called the Bondi mass aspect, the angular momentum aspect and the shear tensor, respectively[3]. The shear tensor satisfies $\gamma^{AB}C_{AB} = 0$ as part of

---

[3]The definition of the angular momentum aspect varies in the literature. The conventions adopted by Flanagan–Nichols [44] or by Barnich–Troessaert and Compère–Fiorucci–Ruzziconi [43, 45, 46] are related to

the gauge conditions. Einstein's equations also imply the evolution equations

$$\partial_u m = -\frac{1}{8} N_{AB} N^{AB} + \frac{1}{4} \nabla_A \nabla_B N^{AB} - 4\pi \lim_{r \to \infty} \left( r^2 T_{uu} \right), \tag{2.3a}$$

$$\partial_u N_A = -u\, \partial_A \partial_u m + \frac{1}{4} \partial_A \left( N_{BC} C^{BC} \right) - \frac{1}{4} \nabla_B \left( C^{BC} N_{CA} \right) + \frac{1}{2} C_{AB} \nabla_C N^{BC} \tag{2.3b}$$
$$- \frac{1}{4} \nabla_B \left( \nabla^B \nabla^C C_{AC} - \nabla_A \nabla_C C^{BC} \right) - 8\pi \lim_{r \to \infty} \left( r^2 T_{uA} \right),$$

where $T_{\mu\nu}$ is the matter stress tensor and $N_{AB} \equiv \partial_u C_{AB}$ is the News tensor.

To proceed in the analysis of the matching with spacelike infinity we assume that this gauge is well-suited to describe the region $\mathscr{I}^+_-$ in the limit $u \to -\infty$ and we specify the following falloff conditions compatible with the evolution equations

$$m = m^0 + u^{-1} m^1 + o(u^{-1}), \tag{2.4a}$$

$$N_A = N_A^0 + o(u^0), \tag{2.4b}$$

$$C_{AB} = C_{AB}^0 + u^{-1} C_{AB}^1 + o(u^{-1}), \tag{2.4c}$$

together with the falloff rate of the matter stress tensor near $\mathscr{I}^+_-$,

$$\lim_{r \to \infty} r^2 T_{uu} = o(u^{-2}), \qquad \lim_{r \to \infty} r^2 T_{uA} = o(u^{-1}). \tag{2.5}$$

Such conditions corresponds to those typically underlying the proofs of the relation between soft theorems and asymptotic symmetries. They are inspired by those resulting from the well-posed Cauchy problem studied by Christodoulou and Klainerman (CK), although here $m^0$ is not restricted to be a constant and $N_{AB}$ falls off faster than the $O(u^{-\frac{3}{2}})$ obtained by CK [17]. Refer to Section 7 for further important comments on such falloff behaviour. For the rest of our arguments, we need to notice that the evolution equation (2.3a)-(2.3b) directly imply

$$\nabla^B \left( \nabla_B \nabla^C C_{AC}^0 - \nabla_A \nabla^C C_{BC}^0 \right) = 0, \tag{2.6}$$

the one used here, respectively by

$$N_A^{\mathrm{FN}} = N_A + u\, \partial_A m,$$

$$N_A^{\mathrm{BT}} = N_A + u\, \partial_A m - \frac{3}{32} \partial_A (C_{BC} C^{BC}) - \frac{1}{4} C_{AB} \nabla_C C^{BC}.$$

The quantity $\phi$ is needed for completeness of the map between Bondi and Beig-Schmidt gauges at the order we work. The only information we actually use is that is behaves like $\phi = u\, \phi^{-1} + \phi^0 + o(u^0)$. The reader can find its explicit expression in appendix B.

and

$$m^1 = \frac{1}{4}\nabla^A\nabla^B C^1_{AB}\,, \tag{2.7a}$$

$$\partial_A m^1 = \frac{1}{4}\nabla^B\left(\nabla_B\nabla^C C^1_{AC} - \nabla_A\nabla^C C^1_{BC}\right)\,. \tag{2.7b}$$

The meaning of these constraints becomes manifest once we decompose $C^0_{AB}$ and $C^1_{AB}$ in electric and magnetic parts,

$$C^i_{AB} = -2\nabla_A\nabla_B C^i + \gamma_{AB}\nabla^2 C^i + \epsilon_{C(A}\nabla_{B)}\nabla^C\Psi^i\,, \qquad i = 0,1\,, \tag{2.8}$$

where $C^i$ is the corresponding electric scalar potential and $\Psi^i$ is the corresponding magnetic pseudo-scalar potential. Note that the $l = 0,1$ spherical harmonics in $C^i$ and $\Psi^i$ do not contribute to (2.8). We can check that the two differential operators appearing in (2.6)-(2.7) respectively project out the electric or magnetic modes,

$$\nabla^A\nabla^B C^i_{AB} = -\nabla^2(\nabla^2 + 2)C^i\,, \tag{2.9a}$$

$$\nabla^B\left(\nabla_B\nabla^C C^i_{AC} - \nabla_A\nabla^C C^i_{BC}\right) = -\epsilon_{AB}\nabla^B\nabla^2(\nabla^2 + 2)\Psi^i\,. \tag{2.9b}$$

Hence the constraint (2.6) requires $\nabla^2(\nabla^2 + 2)\Psi^0 = 0$ which eliminates spherical harmonics with $l > 1$ in $\Psi^0$. Since $C^1$ and $\Psi^1$ are necessarily independent, the constraints (2.7) can only be satisfied provided $m^1 = \nabla^2(\nabla^2 + 2)C^1 = \nabla^2(\nabla^2 + 2)\Psi^1 = 0$ which similarly eliminate all spherical harmonics with $l > 1$ in $C^1$ and $\Psi^1$. In summary, we can write

$$C^0_{AB} = -2\nabla_A\nabla_B C + \gamma_{AB}\nabla^2 C\,, \qquad m^1 = C^1_{AB} = 0\,, \tag{2.10}$$

where the electric potential $C \equiv C^0$ is known as the supertranslation Goldstone mode. In particular we conclude that the News tensor satisfies the stronger falloff $N_{AB} = o(u^{-2})$. This stronger falloff is in fact required for finiteness of the BMS charge fluxes along $\mathscr{I}$ [47], and enters the assumptions for deriving the subleading soft graviton theorem [49].

# 3  Gravity at spatial infinity

The connection between quantities at $\mathscr{I}^-_+$ and $\mathscr{I}^+_-$ will involve the dynamics of the gravitational field near spatial infinity $i^0$. A convenient way to describe this dynamics is to adopt the Beig–Schmidt gauge [21]

$$ds^2 = N^2\,d\rho^2 + H_{ab}\left(N^a\,d\rho + dx^a\right)\left(N^b\,d\rho + dx^b\right)\,, \tag{3.1}$$

where spatial infinity is approached in the limit $\rho \to \infty$. As this limit is taken, the metric behaves as

$$N = 1 + \frac{\sigma}{\rho} \,, \tag{3.2a}$$

$$H_{ab}N^b = o(\rho^{-1}) \,, \tag{3.2b}$$

$$H_{ab} = \rho^2 \left( h_{ab} + \rho^{-1}f_{ab} + \frac{\log \rho}{\rho^2} i_{ab} + \rho^{-2}j_{ab} + o(\rho^{-2}) \right) \,. \tag{3.2c}$$

We assume sufficient falloff of the matter stress tensor ensuring that it does not affect the dynamics of the quantities introduced in (3.2),

$$T_{\rho\rho} = o(\rho^{-4}) \,, \qquad T_{\rho a} = o(\rho^{-3}) \,, \qquad T_{ab} = o(\rho^{-2}) \,. \tag{3.3}$$

Einstein equations at leading order imply

$$R[h]_{ab} = 2h_{ab} \,. \tag{3.4}$$

We take $h_{ab}$ to be globally the three-dimensional de Sitter space $\mathcal{H}$. We stick to the boundary conditions in [27], where $h_{ab}$ is not allowed to fluctuate. We treat $h_{ab}$ as a genuine metric on $\mathcal{H}$ and define the corresponding covariant derivative $D_a$. All three-dimensional indices $a, b, c, ...$ are raised and lowered with this metric.

The leading non-vanishing terms of the electric and magnetic parts of the Weyl tensor are respectively

$$E_{ab} = - \left( D_a D_b + h_{ab} \right) \sigma \,, \qquad B_{ab} = \frac{1}{2} \epsilon_a{}^{cd} D_c k_{db} \,, \tag{3.5}$$

where

$$k_{ab} \equiv f_{ab} + 2\sigma h_{ab} \,. \tag{3.6}$$

The fields $\sigma$ and $k_{ab}$ play the role of potentials for the two components of the Weyl tensor. In order to allow for a well-posed action principle, the trace of $k_{ab}$ must vanish as part of the boundary conditions [27],

$$k_a^a = h^{ab}k_{ab} = 0 \,. \tag{3.7}$$

Given these definitions and boundary conditions, Einstein's equations reduce to dynamical equations on the three-dimensional de Sitter hyperboloid $\mathcal{H}$, which can be solved order by order in the $\rho^{-1}$ expansion. The leading order fields $\sigma, k_{ab}$ and $i_{ab}$ satisfy homogeneous partial differential equations, and act as sources for the subleading field $j_{ab}$. More specifically, the homogeneous equations satisfied by the leading fields are given by

$$(D^2 + 3)\sigma = 0 \,, \qquad (D^2 - 3)k_{ab} = 0 \,, \qquad D^a k_{ab} = 0 \,, \tag{3.8}$$

and

$$(D^2 - 2)i_{ab} = 0, \qquad i^a_a = D^a i_{ab} = 0. \tag{3.9}$$

The inhomogeneous equation satisfied by the subleading field $j_{ab}$ is given by

$$\left(D^2 - 2\right) j_{ab} = 2i_{ab} + S_{ab}, \tag{3.10}$$

subject to the constraints

$$j^a_a = 12\sigma^2 + D_a\sigma\, D^a\sigma + \frac{1}{4}k^{ab}k_{ab} + k^{ab}D_aD_b\sigma, \tag{3.11a}$$

$$D^b j_{ba} = \frac{1}{2}k^c_b\, D^b k_{ca} + D_a\left(8\sigma^2 + D_a\sigma\, D^a\sigma - \frac{1}{8}k^{cd}k_{cd} + k^{cd}D_cD_d\sigma\right). \tag{3.11b}$$

The constraints (3.11) determine the trace and divergence of $j_{ab}$ in terms of the first order data $\sigma$ and $k_{ab}$. The remaining equation (3.10) should be understood as a proper hyperbolic equation for the remaining undetermined degrees of freedom carried by $j_{ab}$[4]. This evolution equation is sourced by a term $S_{ab}$ quadratic in $\sigma$ and $k_{ab}$,

$$S_{ab} = \mathrm{NL}_{ab}(\sigma, \sigma) + \mathrm{NL}_{ab}(\sigma, k) + \mathrm{NL}_{ab}(k, k), \tag{3.12}$$

which is the manifestation of the nonlinear nature of Einstein's equations. We refer to appendix C of [27] for explicit expression of (3.12) which is lengthy but not particularly illuminating.

One important goal of this work is to connect fields near $i^0$ in Beig–Schmidt gauge to fields near $\mathscr{I}^\pm$ in Bondi gauge. For that purpose we now study the behavior of the fields $\sigma$, $k_{ab}$ and $j_{ab}$ in the limits to infinite past and future on the hyperboloid $\mathcal{H}$, which we can expect to connect to past and future null infinity $\mathscr{I}^\pm$ respectively. It will not be required to discuss $i_{ab}$ further since we will find in section 4 that no such term is needed in order to account for the Bondi data appropriate to the gravitational scattering problem. Covering $\mathcal{H}$ with coordinates $(\tau, x^A)$ and metric

$$ds^2_{\mathcal{H}} = -d\tau^2 + \cosh^2\tau\, \gamma_{AB}\, dx^A\, dx^B, \tag{3.13}$$

the loci of interest correspond to the limits $\tau \to \pm\infty$. For simplicity we will only describe the late-time limit and expand the fields $\sigma$, $k_{ab}$ and $j_{ab}$ in the small parameter $e^{-\tau}$, but a similar expansion in the early-time limit obviously holds. Such expansions are completely analogous to the usual Fefferman–Graham expansions in anti-de Sitter space. We give the details of these computations in appendix A and collect the relevant results here.

---

[4]Compère and Dehouck made this point more manifest by rewriting (3.10) as hyperbolic equation for a tracefree and divergencefree tensor $V_{ab}$ [27], building on the seminal work of Beig and Schmidt [21].

**Leading fields.** The large-$\tau$ expansion of the electric potential $\sigma$ and magnetic potential $k_{ab}$ are found to be

$$\sigma(\tau, x) = e^\tau \, \sigma^{(-1)} + e^{-\tau} \sigma^{(1)} + e^{-3\tau} \tau \, \tilde\sigma + e^{-3\tau} \, \sigma^{(3)} + \dots, \tag{3.14}$$

and

$$k_{\tau\tau} = e^{-3\tau} \tau \, \tilde k_{\tau\tau} + e^{-3\tau} \, k_{\tau\tau}^{(3)} + \dots, \tag{3.15a}$$

$$k_{\tau A} = e^{-\tau} \tau \, \tilde k_{\tau A} + e^{-\tau} \, k_{\tau A}^{(1)} + \dots, \tag{3.15b}$$

$$k_{AB} = e^\tau \tau \, \tilde k_{AB} + e^\tau \, k_{AB}^{(-1)} + \dots. \tag{3.15c}$$

The equations of motion (3.8) are quadratic differential equations, and as such they generically admit two independent sets of solutions with distinct asymptotic behaviors. The two independent solutions for the electric potential $\sigma$ are characterized respectively by $\sigma^{(-1)}$ and $\sigma^{(3)}$, while all the other functions appearing in the $\tau$-expansion (3.14) can be fully determined in terms of these data. A similar structure applies to the components of $k_{ab}$, where the two sets of independent solutions are characterized by the first two functions on the sphere appearing in each of the $\tau$-expansions (3.15).

**Subleading field.** The analysis of the large-$\tau$ behavior of $j_{ab}$ is significantly more delicate due to the appearance of terms quadratic in $\sigma$ and $k_{ab}$ on the right-hand side of (3.10)-(3.11). These terms are the manifestation of the nonlinear nature of Einstein's equations, and their careful treatment is precisely what will allow us to prove the antipodal matching condition of the angular momentum aspect without imposing dramatic restrictions on the Bondi data. We can summarise the situation in the following way. The solutions to (3.10)-(3.11) are given by the superposition of a particular solution that depends on pre-determined source terms such as $S_{ab}$, and a combination of homogeneous solutions. The asymptotic behavior of the homogeneous solutions is easily worked out,

$$j_{\tau\tau} = e^{-2\tau} j_{\tau\tau}^{(2)} + e^{-4\tau} j_{\tau\tau}^{(4)} + \dots, \tag{3.16a}$$

$$j_{\tau A} = j_{\tau A}^{(0)} + e^{-2\tau} j_{\tau A}^{(2)} + \dots, \tag{3.16b}$$

$$j_{AB} = e^{2\tau} j_{AB}^{(-2)} + j_{AB}^{(0)} + \dots, \tag{3.16c}$$

while the behavior of the particular solution strongly depends on the form of $\sigma$ and $k_{ab}$. A key result of the analysis to be presented in section 4 is that the Bondi data maps onto a *subset* of the allowed Beig–Schmidt data, in such a way that the large-$\tau$ behavior of the particular solution is subleading compared to that of the homogeneous solutions. Thus

(3.16) holds true for the full solution of Einstein's equations provided such a solution can be mapped onto the Bondi phase space. On the other hand, a generic solution of the Beig–Schmidt equations (3.10)-(3.11) which is not connected to the Bondi phase space would see its leading asymptotics (3.16) modified due to weaker falloffs of the source terms quadratic in $\sigma$ and $k_{ab}$. This clean separation between large-$\tau$ asymptotics of the homogeneous and particular solutions is a property of the Bondi phase space, not one of the larger Beig–Schmidt phase space. As we will further show in section 4, the angular momentum aspect sits in $j_{\tau A}^{(2)}$ and assessing that this term is fully governed by a homogeneous solution appears crucial to the derivation of the corresponding antipodal matching condition in section 5.

# 4    From Bondi to Beig-Schmidt

At leading order in $r$ and $\rho$ respectively, the Bondi and Beig-Schmidt metrics are simply that of Minkowski space written in two different coordinate systems. The coordinate transformation between these two is explicitly given by

$$u = -\rho\, e^{-\tau}, \tag{4.1a}$$

$$r = \rho \cosh \tau. \tag{4.1b}$$

Obviously there exists an analogous coordinate transformation to the advanced Bondi gauge describing the neighborhood of $\mathscr{I}^-$,

$$v = \rho\, e^{\tau}, \tag{4.2a}$$

$$r = \rho \cosh \tau. \tag{4.2b}$$

We want to find a map between data of asymptotically flat gravity at null and spatial infinity. We will proceed by explicit coordinate transformation from the Bondi gauge to the Beig–Schmidt gauge. However each of these two asymptotic expansions are valid in different regions of spacetime and one can only hope to relate them where these expansions overlap. This happens in the regime $r \to \infty$, $u \to -\infty$ ($v \to \infty$) or equivalently in the limit $\rho, \tau \to \infty$ ($\tau \to -\infty$), which can intuitively be thought of as the neighborhood of $\mathscr{I}_-^+$ ($\mathscr{I}_+^-$). To be more precise, we will start from Bondi metrics written as a double asymptotic expansion in $r \gg |u| \gg 1$, which we will map to Beig–Schmidt metrics written as a double asymptotic expansion in $\rho \gg e^{\tau} \gg 1$. Since

$$\frac{u}{r} = O(e^{-2\tau}), \tag{4.3}$$

terms that are subleading in $r$ but overleading in $u$ will contribute at the same order in $\rho$ but to subleading order in $e^{-\tau}$. The explicit details of this transformation are relegated to appendix B.

A first important observation is that the logarithmic term $i_{ab}$ is not generated by this mapping of the Bondi data onto the Beig–Schmidt data. For the electric potential, the map yields

$$\sigma^{(-1)} = \sigma^{(1)} = \tilde{\sigma} = 0 \,, \qquad \sigma^{(3)} = 2m^0 \,, \tag{4.4}$$

while for the magnetic potential, we find

$$\tilde{k}_{\tau A} = \tilde{k}_{AB} = 0 \,, \qquad k_{\tau A}^{(1)} = 2\nabla^B C_{AB}^0 \,, \qquad k_{AB}^{(-1)} = \frac{1}{2}\, C_{AB}^0 \,. \tag{4.5}$$

Note that this is enough to also determine $\tilde{k}_{\tau\tau}$ from the the constraints $k_a^a = D^a k_{ab} = 0$, and the only undetermined data is therefore $k_{\tau\tau}^{(3)}$. We show in appendix C that (4.5) necessarily implies that $k_{ab}$ takes the form

$$k_{ab} = -\left(D_a D_b + h_{ab}\right)\Phi \,, \qquad \left(D^2 + 3\right)\Phi = 0 \,, \tag{4.6}$$

where $\Phi$ is the Goldstone mode of *Spi-supertranslations* [50]. Just like the electric potential, $\Phi$ is fully characterised by its leading asymptotic data $\Phi^{(-1)}$ and $\Phi^{(3)}$. In appendix C we confirm the identification $\Phi^{(-1)} = C$ with the supertranslation mode (2.10) previously made in [50]. The remaining degree of freedom $\Phi^{(3)}$ then corresponds to the undetermined data $k_{\tau\tau}^{(3)}$. It is known since the work of Troessaert that $\Phi^{(3)}$ is in fact pure gauge [39], and we can therefore consider $\Phi^{(3)} = k_{\tau\tau}^{(3)} = 0$ without loss of generality. Thus $k_{ab}$ is fully determined by the supertranslation mode $C$. Another direct consequence of the restricted form (4.6) is the vanishing of the leading magnetic Weyl tensor (3.5),

$$B_{ab} = 0 \,. \tag{4.7}$$

This result has often been assumed in the literature [25, 26, 51], and played a crucial role in the previous matching of Lorentz charges by Prabhu and Shehzad [24]. We just showed that (4.7) actually follows from the Bondi phase space described in section 2 and considered appropriate to the gravitational scattering problem.

Similarly, we find that the leading asymptotic data allowed by the homogeneous solutions for the subleading field $j_{ab}$ actually vanishes,

$$j_{\tau\tau}^{(2)} = j_{\tau A}^{(0)} = j_{AB}^{(-2)} = 0 \,, \tag{4.8}$$

while the subleading asymptotic data is given by

$$j_{\tau\tau}^{(4)} = 4\nabla_A C_{BC}^0 \nabla^A C_0^{BC} - 4\nabla_E C_{AB}^0 \nabla^A C_0^{EB} + 64\phi^0 \,, \tag{4.9a}$$

$$j_{\tau A}^{(2)} = 4N_A^0 + C_{AB}^0 \nabla_C C_0^{BC} \,, \tag{4.9b}$$

$$j_{AB}^{(0)} = \frac{1}{8} C_{CD}^0 C_0^{CD} \gamma_{AB} \,. \tag{4.9c}$$

At this point we can compute the trace and divergence of $j_{ab}$ and verify that they do satisfy the constraints (3.11) resulting from Einstein's equations in Beig–Schmidt gauge. Up to the available orders in $\tau$, we indeed find perfect agreement between the direct computation from (4.9) and the evaluation of (3.11) requiring only knowledge of the leading data (4.4)-(4.5), namely

$$j_a^a = e^{-2\tau} C_{AB}^0 C_0^{AB} + O(e^{-4\tau}) \,, \tag{4.10a}$$

$$D^b j_{b\tau} = -e^{-2\tau} C_{AB}^0 C_0^{AB} + O(e^{-4\tau}) \,, \tag{4.10b}$$

$$D^b j_{bA} = \frac{1}{2} e^{-2\tau} \partial_A \left( C_{BC}^0 C_0^{BC} \right) + O(e^{-4\tau}) \,. \tag{4.10c}$$

We can now come back to the discussion started at the end of section 3 regarding the clean separation observed between homogeneous and particular solutions of $j_{ab}$. By explicit coordinate transformation between Bondi and Beig–Schmidt gauges, we just obtained a large-$\tau$ behavior for $j_{ab}$ which coincides with that of the homogeneous solutions to (3.10)-(3.11) and described in (3.16). Consistency of our findings with the Beig–Schmidt dynamics therefore requires that the particular solution of $j_{ab}$ determined by the source terms in (3.10) be subleading in $\tau$, a fact which we have verified in appendix A by direct evaluation of the source terms. Thus the quantities (4.9) are entirely governed by homogeneous solutions to (3.10)-(3.11), which will prove crucial to the derivation of the antipodal matching condition of the angular momentum aspect $N_A$.

The map between data at $\mathscr{I}_+^-$ in retarded Bondi gauge and data in the limit $\tau \to -\infty$ in Beig–Schmidt gauge is worked out in a similar way. The resulting identifications have the same functional form as above, with a few minus signs differences. The rule of thumb is that any $\tau$ index yields a relative minus sign. In particular, we find

$$\sigma(\tau, x^A) = 2m^0\big|_{\mathscr{I}_+^-} e^{3\tau} + O(e^{5\tau}) \,, \tag{4.11a}$$

$$k_{AB}(\tau, x^A) = \frac{1}{2} C_{AB}^0\big|_{\mathscr{I}_+^-} e^{-\tau} + O(e^{\tau}) \,, \tag{4.11b}$$

$$j_{\tau A}(\tau, x^A) = - \left(4N_A^0 + C_{AB}^0 \nabla_C C_0^{BC}\right)\big|_{\mathscr{I}_+^-} e^{2\tau} + O(e^{4\tau}) \,. \tag{4.11c}$$

The antipodal matching conditions, to be studied in the next section, give relations between quantities defined in the two limits $\tau \to \infty$ and $\tau \to -\infty$.

# 5    Antipodal matching relations

We are now ready to give a complete derivation of the antipodal matching conditions used by Strominger and crucial in establishing an equivalence between soft graviton theorems and Ward identities associated with BMS symmetries [4, 5, 52]. These antipodal matching relations are

$$\Upsilon^* m|_{\mathscr{I}^+_-} = m|_{\mathscr{I}^-_+} \, , \tag{5.1a}$$

$$\Upsilon^* C_{AB}|_{\mathscr{I}^+_-} = - C_{AB}|_{\mathscr{I}^-_+} \, , \tag{5.1b}$$

$$\Upsilon^* N_A|_{\mathscr{I}^+_-} = N_A|_{\mathscr{I}^-_+} \, . \tag{5.1c}$$

In the previous section we mapped the leading Bondi data onto a subset of the leading Beig–Schmidt data. A key observation is that the latter is fully governed by homogeneous solutions of the Beig–Schmidt equations. In this section we show that the antipodal matching relations follow from the parity properties of these homogeneous solutions under the $\mathcal{H}$-antipodal map $\Upsilon_{\mathcal{H}}$.

As commented in the introduction, various works already exist on this topic [23,24,39,42]. As summarised in [42], in the Hamiltonian framework conditions have been given at spacelike infinity in order to recover BMS symmetries and argue in favour of the antipodal matching among past and future null infinities. Earlier, some steps towards the derivation of (5.1) were taken by Troessaert [39] by mapping the Bondi-Sachs gauge to the Beig-Schmidt gauge at leading order. After showing that the Lie algebras associated with global BMS symmetries ($\mathscr{I}$) and Spi-symmetries ($i^0$) are isomorphic, he argued that the charge density and symmetry parameter associated with Spi-supertranslations both satisfy antipodal relations. However his analysis was restricted to linearized gravity. Recently Prabhu [23] and Prabhu and Shehzad [24] tackled the antipodal matching of both supertranslation and angular momentum charges using the formalism of Ashtekar and Hansen [25], formally without restricting to the linear theory. Specifically, a number of assumptions were required to achieve the angular momentum matching, among them the vanishing of the leading magnetic Weyl tensor $B_{ab}$ was assumed and the inhomogeneous terms were neglected in the subleading equation of motion defining the angular momentum data [24]. In this approach, furthermore, the matching is somewhat indirect because the charges at $\mathscr{I}^+$ and $i^0$ are independent and linked through the observation that the asymptotic limit of certain bulk spacetime quantity is the same as the limit toward $i^0$ ($\mathscr{I}^+$) of the charge defined on $\mathscr{I}^+$ ($i^0$).

The analysis that we present here completes these results in various important ways. First, the connection we make between $\mathscr{I}$ and $i^0$ goes beyond that of matching Lie algebras

associated with asymptotic symmetries, since we have provided in section 4 a precise dictionary between Bondi data and Beig–Schmidt data. This allows us to unambiguously identify where the initial data of the shear $C_{AB}$, mass aspect $m$ and angular momentum aspect $N_A$ sits in the Beig–Schmidt gauge, and to proceed with the study of their parity properties and matching of the asymptotic charges at $\mathscr{I}$ and $i^0$.

Such analysis does not require any further assumption in the bulk, except those made on the structure near null infinity. For example, as seen in the previous section, the leading magnetic Weyl tensor at spacelike infinity $B_{ab}$ vanishes as a consequence of these. In the current section, the key point we will use in the derivation of the antipodal matching (5.1) also follows from the structure at null infinity. The relations (5.1) in fact stem from parity properties of the *homogeneous solutions* for $\sigma, k_{ab}$ and $j_{ab}$ under the $\mathcal{H}$-antipodal map $\Upsilon_{\mathcal{H}}$. While these were crucially used in [23, 24, 39] as well, here we do not need to discard source terms quadratic in $\sigma$ and $k_{ab}$ in the equations (3.10)-(3.11) determining $j_{ab}$. Rather, the dictionary of section 4 between Bondi and Beig–Schmidt data shows that such source terms do not affect the large-$\tau$ behavior of $j_{ab}$, and hence the angular momentum aspect $N_A$ is still fully governed by homogeneous solutions of $j_{ab}$. The derivation of the antipodal relations (5.1) based on parity properties of homogeneous solutions to the Beig–Schmidt equations then proceeds unobstructed.

**Harmonic and Legendre functions.** The dependence on the sphere coordinates $x^A$ will be treated by decomposition into scalar spherical harmonics $Y_l^m(x^A)$, satisfying

$$\nabla^2 Y_l^m = -l(l+1)Y_l^m\,, \qquad \Upsilon^* Y_l^m = (-1)^l\, Y_l^m\,. \tag{5.2}$$

Beig–Schmidt equations then generically reduce to Legendre equations of the form

$$\left[(1-s^2)\partial_s^2 - 2s\partial_s + l(l+1) - \frac{n^2}{1-s^2}\right] F(s) = S(s)\,, \quad s \equiv \tanh\tau \in (-1,1)\,, \tag{5.3}$$

with $S(s)$ a generic source term. The homogeneous solutions are associated Legendre functions on the cut [53],

$$P_l^n(s)\,, Q_l^n(s)\,, \qquad (n \geq 0)\,, \tag{5.4}$$

satisfying the parity properties

$$P_l^n(-s) = (-1)^{l+n}P_l^n(s)\,, \qquad Q_l^n(-s) = (-1)^{l+n+1}Q_l^n(s)\,. \tag{5.5}$$

Of importance to us will be their asymptotic behavior in the limit $s \to \pm 1$, or equivalently in the limit $\tau \to \pm\infty$. For $l \geq n$, we have

$$P_l^n(s) = O\left((1-s)^{n/2}\right) = O(e^{-n\tau})\,, \qquad Q_l^n(s) = O\left((1-s)^{-n/2}\right) = O(e^{n\tau})\,, \tag{5.6}$$

while solutions with $l < n$ have a separate asymptotic behavior. For $n = 1$, we have

$$P_0^1(s), Q_0^1(s) = O(1/\sqrt{1-s}) = O(e^\tau), \tag{5.7}$$

while for $n = 2$, we have

$$P_0^2(s), P_1^2(s), Q_0^2(s), Q_1^2(s) = O\left(1/(1-s)\right) = O(e^{2\tau}). \tag{5.8}$$

We can now proceed to the derivation of the antipodal relations (5.1).

**Mass aspect.** The initial value of the Bondi mass aspect at $\mathscr{I}_-^+$ is carried by

$$\sigma^{(3)} = 2m\big|_{\mathscr{I}_-^+}, \tag{5.9}$$

where the electric potential $\sigma$ solves the homogeneous equation

$$\left[-\partial_\tau^2 - 2\tanh\tau\,\partial_\tau + \cosh^{-2}\tau\,\nabla^2 + 3\right]\sigma = 0. \tag{5.10}$$

We introduce the variable $s = \tanh\tau \in (-1, 1)$ and decompose $\sigma$ in spherical harmonics,

$$\sigma(s, x^A) = \sqrt{1-s^2}\sum_{l,m}\sigma_{lm}(s)Y_l^m(x^A). \tag{5.11}$$

The coefficients then satisfy the Legendre differential equation

$$\left[(1-s^2)\partial_s^2 - 2s\partial_s + l(l+1) - \frac{4}{1-s^2}\right]\sigma_{lm}(s) = 0, \tag{5.12}$$

whose solutions are the Legendre functions

$$P_l^2(s), \, Q_l^2(s). \tag{5.13}$$

Taking into account the prefactor $\sqrt{1-s^2}$ in (5.11), we can confirm that independent solutions behave either as $O(e^\tau)$ or $O(e^{-3\tau})$ in agreement with (3.14). In section 4 we found that the mode $O(e^\tau)$ is absent, by explicit mapping of the Bondi data onto the Beig-Schmidt data. Therefore we conclude that the relevant general solution for the electric potential takes the form

$$\sigma(s, x^A) = \sqrt{1-s^2}\sum_{l=2}^\infty\sum_{m=-l}^l a_{lm}P_l^2(s)Y_l^m(x^A). \tag{5.14}$$

In particular, it is parity-even under the $\mathcal{H}$-antipodal map,

$$\Upsilon_{\mathcal{H}}^*\sigma = \sigma, \tag{5.15}$$

in agreement with earlier discussions on the existence of a regular null infinity [23,38,39,50]. Making use of (4.11), this also directly yields the antipodal relation for the Bondi mass aspect,

$$\Upsilon^*m\big|_{\mathscr{I}_-^+} = m\big|_{\mathscr{I}_+^-}. \tag{5.16}$$

**Shear tensor.** The initial value of the shear tensor at $\mathscr{I}_-^+$ is encoded in the leading nontrivial component of the magnetic potential,

$$2\,k_{AB}^{(-1)} = C_{AB}\big|_{\mathscr{I}_-^+} = -2\nabla_A\nabla_B C + \gamma_{AB}\nabla^2 C\,. \tag{5.17}$$

We know from appendix C that the relevant solutions for $k_{ab}$ take the form

$$k_{ab} = -\left(D_a D_b + h_{ab}\right)\Phi\,, \qquad \left(D^2 + 3\right)\Phi = 0\,, \tag{5.18}$$

where the supertranslation mode $C$ is identified with the leading large-$\tau$ behavior of $\Phi$,

$$\Phi(\tau, x^A) = e^\tau C(x^A) + O(e^{-\tau})\,. \tag{5.19}$$

Thus it is the scalar potential $\Phi$ that carries the relevant information. Its evolution equation is identical to that of $\sigma$, although this time there is no restriction on its asymptotic behavior. Given (5.19) we are interested in the set of solutions scaling like $O(e^\tau)$,

$$\Phi(s, x^A) = \sqrt{1 - s^2}\left(\sum_{l=0,1}\sum_{m=-l}^{l} a_{lm}\, P_l^2(s) + \sum_{l=0}^{\infty}\sum_{m=-l}^{l} b_{lm}\, Q_l^2(s)\right)Y_l^m(x^A)\,. \tag{5.20}$$

This solution *almost* has definite odd parity under $\Upsilon_\mathcal{H}$, which is however spoiled by the $a_{lm}$ with $l = 0, 1$. But these do not contribute to the shear (5.17) since the four lowest spherical harmonics $Y_0^m, Y_1^m$ are precisely annihilated by the differential operator $-2\nabla_A\nabla_B + \gamma_{AB}\nabla^2$. Moreover it can be seen from (C.6) that none of the data specifying $k_{ab}$ is actually sensitive to the $a_{lm}$, such that we can as well set them to zero. This implies that the $k_{ab}$ satisfies the parity property

$$\Upsilon_\mathcal{H}^* k_{ab} = -k_{ab}\,. \tag{5.21}$$

Strictly speaking this requires the solutions of $\Phi$ scaling like $O(e^{-3\tau})$ to be absent, and this can always be achieved without loss of generality since these solutions can be removed by pure gauge transformations [39]. It is sometimes useful to express these relations in terms of $k_{\tau\tau}$, $k_{\tau A}$ and $k_{AB}$ viewed as time-dependent scalar, vector and tensor fields on $\mathbb{S}^2$, respectively. These read

$$\Upsilon^* k_{\tau\tau}(-\tau) = -k_{\tau\tau}(\tau)\,, \qquad \Upsilon^* k_{\tau A}(-\tau) = k_{\tau A}(\tau)\,, \qquad \Upsilon^* k_{AB}(-\tau) = -k_{AB}(\tau)\,. \tag{5.22}$$

Using (4.11), this yields in particular the antipodal relation of the shear tensor,

$$\Upsilon^* C_{AB}\big|_{\mathscr{I}_-^+} = -\,C_{AB}\big|_{\mathscr{I}_+^-}\,. \tag{5.23}$$

**Angular momentum aspect.** The initial value of the angular momentum aspect at $\mathscr{I}_-^+$ is carried by the leading data of the field $j_{ab}$, namely

$$j_{\tau A}^{(2)} = 4 N_A + C_{AB} \nabla_C C^{BC} \big|_{\mathscr{I}_-^+} \,, \tag{5.24}$$

and the relevant homogeneous solutions satisfy

$$\left[ -\partial_\tau^2 - 2 \tanh \tau \, \partial_\tau + \cosh^{-2} \tau \, (\nabla^2 - 1) \right] j_{\tau A} = 0 \,. \tag{5.25}$$

We make use of Helmholtz decomposition

$$j_{\tau A} = \nabla_A \Psi_1 + \epsilon_{AB} \nabla^B \Psi_2 \,, \tag{5.26}$$

where $\Psi_1$ is a scalar and $\Psi_2$ is a pseudo-scalar under the antipodal map $\Upsilon$, and further decompose them in spherical harmonics,

$$\Psi_i(s, x^A) = \sqrt{1 - s^2} \sum_{l,m} \Psi_{i,lm}(s) \, Y_l^m(x^A) \,, \qquad i = 1, 2 \,, \tag{5.27}$$

such that we end up with the ordinary differential equations

$$\left[ (1 - s^2) \partial_s^2 - 2 s \partial_s + l(l+1) - \frac{1}{1 - s^2} \right] \Psi_{i,lm}(s) = 0 \,, \qquad i = 1, 2 \,. \tag{5.28}$$

The solutions with asymptotic behavior $O(e^{-2\tau})$ are of the form

$$\Psi_i(s, x^A) = \sqrt{1 - s^2} \sum_{l=1}^{\infty} \sum_{m=-l}^{l} a_{i,lm} \, P_l^1(s) \, Y_l^m(x^A) \,. \tag{5.29}$$

These are odd under $\Upsilon_{\mathcal{H}}$ following the parity properties of spherical harmonics and Legendre functions. Hence the corresponding homogeneous solution $j_{\tau A}$, viewed as a time-dependent vector field on the sphere $\mathbb{S}^2$, satisfies

$$\Upsilon^* j_{\tau A}(-\tau) = -j_{\tau A}(\tau) \,. \tag{5.30}$$

Making use of (4.11), we thus obtain the antipodal relation of the angular momentum aspect,

$$\Upsilon^* N_A \big|_{\mathscr{I}_-^+} = N_A \big|_{\mathscr{I}_+^-} \,. \tag{5.31}$$

# 6 Matching of BMS charges from $\mathscr{I}$ to $i^0$

As advertised in the introduction we are able to explicitly match various proposals of global[5] BMS charges in Bondi gauge listed in [48] to the charges that were directly constructed in Beig–Schmidt gauge by Compère and Dehouck[6] (CD) [27]. Conservation of the CD charges then directly implies conservation of the corresponding BMS charges across $i^0$.

**BMS charges at $\mathscr{I}_-^+$.**  Global BMS charges are labelled by the quantities $(T, Y^A)$ defined over the celestial sphere $\mathbb{S}^2$, where $T$ is a function parametrising supertranslations while $Y^A$ is a conformal Killing vector parametrising Lorentz rotations and boosts. We give the following parametrisation of the various proposals for global BMS charges in Bondi gauge,

$$Q_{(\alpha,\beta)}[T, Y^A] = \frac{1}{8\pi G} \int_{\mathbb{S}^2} d\Omega \left( 2T\, m + Y^A \hat{N}_A \right), \tag{6.1}$$

with

$$\hat{N}_A \equiv N_A - \frac{\alpha}{16} \partial_A (C_{BC} C^{BC}) - \frac{\alpha}{4} C_{AB} \nabla_C C^{BC} + u\, \frac{\beta}{4} \nabla^B \left( \nabla_B \nabla^C C_{AC} - \nabla_A \nabla^C C_{BC} \right). \tag{6.2}$$

This parametrisation slightly extends the one given in [48] in that we include $\beta$ in order to properly account for the recently constructed charges in [45–47]. Now we evaluate these charges in the limit to $\mathscr{I}_-^+$ ($u \to -\infty$) where we can make use of the expansions (2.4), resulting in

$$\lim_{u \to -\infty} Q_{(\alpha,\beta)}[T] = \frac{1}{4\pi G} \int d\Omega\, T\, m^0, \tag{6.3}$$

$$\lim_{u \to -\infty} Q_{(\alpha,\beta)}[Y^A] = \frac{1}{8\pi G} \int d\Omega\, Y^A N_A^0. \tag{6.4}$$

We note that the parameters $(\alpha, \beta)$ do not actually contribute in this limit. In particular the term controlled by $\alpha$ is proportional to

$$\int d\Omega\, Y^A \left[ \partial_A (C_{BC}^0 C_0^{BC}) + 4 C_{AB}^0 \nabla_C C_0^{BC} \right] = 0, \tag{6.5}$$

---

[5]Charges associated with *generalized* BMS symmetries are also studied in [8, 9, 45–47, 49]. Generalized BMS transformations require to consider generic metrics $\gamma_{AB}$ over the celestial sphere $\mathbb{S}^2$ together with an additional 'conformal' connection $N_{AB}^{\mathrm{vac}}$ also known as tracefree Geroch tensor [49, 54, 55]. When $\gamma_{AB}$ is the unit round sphere metric as it is the case here, one has $N_{AB}^{\mathrm{vac}} = 0$.

[6]They also considered charges associated to so-called *logarithmic translations* [21, 56, 57]. However these asymptotic symmetries generate nonzero $\sigma^{(-1)}$ such that they cannot be considered when the existence of a regular $\mathscr{I}$ is assumed [38, 50].

which integrates to zero on account of the various identities satisfied by $C^0_{AB}$ and $Y^A$.

We show below that the expressions (6.3)-(6.4) indeed coincide the conserved CD charges. Since the latter are conserved, they can be evaluated on any spacelike cut of $\mathcal{H}$ with topology of the sphere $\mathbb{S}^2$. We will restrict to constant-$\tau$ cuts, and subsequently take the limit $\tau \to \infty$ such that we can easily write them in the terms of leading Bondi data $m^0, C^0_{AB}$ and $N^0_A$.

**Supertranslation charges.** The supertranslation charges are given by [29]

$$Q_{\text{CD}}[\omega] = \frac{\cosh^2 \tau}{4\pi G} \oint d\Omega \left( \omega \partial_\tau \sigma - \sigma \partial_\tau \omega \right) , \tag{6.6}$$

where the symmetry parameter $\omega(x^a)$ satisfies the constraint

$$\left( D^2 + 3 \right) \omega = 0 , \tag{6.7}$$

and therefore admits the large-$\tau$ expansion

$$\omega(\tau, x^A) = e^\tau \, \bar\omega(x^A) + O(e^{-\tau}) . \tag{6.8}$$

Under Spi-supertranslation, the magnetic potential $k_{ab}$ and Spi-supertranslation mode $\Phi$ defined in (4.6) transform as

$$\delta_\omega k_{ab} = 2 \left( D_a D_b + h_{ab} \right) \omega , \qquad \delta_\omega \Phi = -2\omega . \tag{6.9}$$

Using (C.3) and (C.8), we can further identify the $\bar\omega$ as the symmetry parameter of supertranslations at $\mathcal{I}$,

$$\bar\omega = -\frac{1}{2} T , \qquad \delta_T C = T . \tag{6.10}$$

By identical arguments as those used in section 5, we also infer the antipodal matching of the supertranslation parameter,[7]

$$\Upsilon^* T \big|_{\mathcal{I}^+_-} = -T \big|_{\mathcal{I}^-_+} . \tag{6.12}$$

This is the condition used by Strominger to single out the diagonal subgroup of $\text{BMS}(\mathcal{I}^+) \times \text{BMS}(\mathcal{I}^-)$ as the symmetry group of the gravitational $\mathcal{S}$-matrix [4]. We evaluate these charges in the limit $\tau \to \infty$ and use the dictionary of section 4 to express them in terms of Bondi data,

$$Q_{\text{CD}}[\omega] = -\frac{1}{4\pi G} \oint d\Omega \, \bar\omega \, \sigma^{(3)} = -\frac{1}{2\pi G} \oint d\Omega \, \bar\omega \, m^0 = \frac{1}{4\pi G} \oint d\Omega \, T \, m^0 . \tag{6.13}$$

This indeed agrees with (6.3).

---

[7]The conventions adopted here are such that the supertranslation mode $C$ transforms in the same way at $\mathcal{I}^\pm$,

$$\delta_T C = T \big|_{\mathcal{I}^+} , \qquad \delta_T C = T \big|_{\mathcal{I}^-} , \tag{6.11}$$

and therefore differ by a relative minus sign from those adopted by Strominger [4].

**Lorentz charges.** The Lorentz charges are given by [29]

$$Q_{\mathrm{CD}}[\xi^a] = \frac{\cosh^2 \tau}{8\pi G} \int d\Omega \, \xi^a \delta_\tau^b \left[ -j_{ab} + \frac{1}{2} i_{ab} + \frac{1}{2} k_a^c k_{cb} + h_{ab} F \right], \tag{6.14}$$

with

$$F \equiv 8\sigma^2 + D^c \sigma D_c \sigma - \frac{1}{8} k^{cd} k_{cd} + k^{cd} D_c D_d \sigma. \tag{6.15}$$

The symmetry parameters $\xi^a$ are the Killing vector fields of the hyperboloid $\mathcal{H}$, and indeed the isometry group of three-dimensional de Sitter space is isomorphic to the Lorentz group $SO(3,1)$. They satisfy

$$D_a \xi_b + D_b \xi_a = 0, \tag{6.16}$$

which in the $(\tau, x^A)$ coordinate system reads

$$0 = \partial_\tau \xi^\tau, \tag{6.17a}$$

$$0 = \partial_\tau \xi_A - \partial_A \xi^\tau - 2 \tanh \tau \, \xi_A, \tag{6.17b}$$

$$0 = \nabla_A \xi_B + \nabla_B \xi_A + 2 \cosh \tau \, \sinh \tau \, \gamma_{AB} \, \xi^\tau. \tag{6.17c}$$

The solutions to these equations are given by

$$\xi^\tau = b(x^A), \tag{6.18a}$$

$$\xi^A = \tilde{\xi}^A + \tanh \tau \, \partial^A b, \qquad \partial_\tau \tilde{\xi}^A = 0, \tag{6.18b}$$

with the constraints

$$0 = \nabla_A \tilde{\xi}_B + \nabla_B \tilde{\xi}_A, \tag{6.19a}$$

$$0 = (\nabla_A \nabla_B + \nabla_B \nabla_A + 2\gamma_{AB}) b. \tag{6.19b}$$

The first constraint implies that $\tilde{\xi}^A$ is a Killing vector field on the sphere $\mathbb{S}^2$ that parametrise rotations. The second constraint implies that $b$ is a linear combination of the three spherical harmonics $Y_{l=1}^m$ parametrising boosts. $\tilde{\xi}^A$ and $b$ have even and odd parities under the antipodal map $\Upsilon$, respectively, such that $\xi^a$ is even under the $\mathcal{H}$-antipodal map $\Upsilon_{\mathcal{H}}$. In the limit $\tau \to \infty$, we can define the vector fields on the sphere

$$Y^A \equiv -\lim_{\tau \to \infty} \xi^A = -(\tilde{\xi}^A + \partial^A b), \qquad b = \frac{1}{2} \nabla_A Y^A. \tag{6.20}$$

Using the above relations, one finds that they satisfy the conformal Killing equation on the sphere,

$$\nabla_A Y_B + \nabla_B Y_A = \gamma_{AB} \nabla_C Y^C. \tag{6.21}$$

Consistently, the group of conformal isometries of $\mathbb{S}^2$ is isomorphic to the Lorentz group SO(3,1). The even parity of $\xi^a$ under $\Upsilon_{\mathcal{H}}$ in particular implies the antipodal matching relation

$$\Upsilon^* Y^A\big|_{\mathscr{I}_-^+} = Y^A\big|_{\mathscr{I}_+^-} . \tag{6.22}$$

We can then evaluate the Lorentz charges (6.14) in the limit $\tau \to \infty$ using the expansions of section 3 and their expressions in terns of Bondi data given in section 4. One obtains

$$Q_{\mathrm{CD}}[\xi^a] = -\frac{1}{32\pi G} \int d\Omega \left[ Y^A \left( -j_{\tau A}^{(2)} + 2k_{AB}^{(-1)} k^{(1)B}{}_\tau \right) - \nabla_A Y^A \, k_{BC}^{(-1)} k^{(-1)BC} \right] \tag{6.23a}$$

$$= \frac{1}{8\pi G} \int d\Omega \, Y^A \left[ N_A^0 - \frac{1}{16} \partial_A (C_{BC}^0 C_0^{BC}) - \frac{1}{4} C_{AB}^0 \nabla_C C_0^{BC} \right] \tag{6.23b}$$

$$= \frac{1}{8\pi G} \int d\Omega \, Y^A N_A^0 \,, \tag{6.23c}$$

where we made use of (6.5) in the last equality. This exactly coincides with (6.4).

**Linearisation stability constraints.** This is a good place to quickly address the integrability conditions presented in [22, 27], also recognised as linearisation stability constraints [58, 59]. These constraints need to be satisfied in order for the solution of the linearised system to lift to a nonlinear solution of Einstein's equations. The integrability conditions relate two types of conserved charges,

$$\tilde{Q}[\xi^a] \equiv \cosh^2 \tau \int_{\mathbb{S}^2} d\Omega \, \xi^a \delta_\tau^b \, \kappa_{ab} \overset{!}{=} -2 \cosh^2 \tau \int_{\mathbb{S}^2} d\Omega \, \xi^a \delta_\tau^b \, i_{ab} \,, \tag{6.24}$$

where the quantity $\kappa_{ab}$ is quadratic in $\sigma$ and $k_{ab}$ and is given explicitly in appendix B of [27]. Our map from Bondi to Beig–Schmidt produced a vanishing $i_{ab}$, and we should therefore ensure that the conserved charge $\tilde{Q}[\xi^a]$ on the left-hand side also vanishes. Taking the limit $\tau \to \infty$, we find

$$\tilde{Q}[\xi^a] = -\frac{1}{4} \int d\Omega \left[ Y^A \partial_A (C_{BC}^0 C_0^{BC}) + \nabla_A Y^A \, C_{BC}^0 C_0^{BC} \right] = 0 \,. \tag{6.25}$$

We conclude that the data which we are considering is that of a proper solution to Einstein's equations.

# 7 Discussion

In this work we derived the antipodal matching relations (5.1) used in proving the equivalence between the (sub)leading soft graviton theorems and (extended) BMS charge conservation

across spatial infinity [4,5,52]. We also explicitly demonstrated that the various proposals for global BMS charges at $\mathscr{I}$ precisely match the conserved charges at spatial infinity. To derive these results we made a few assumptions in section 2 regarding the gravitational phase space at $\mathscr{I}$ in Bondi gauge, in line with what is usually considered appropriate to the gravitational scattering problem (at least within the celestial holography program). We then provided a precise map between gravitational data in Bondi gauge and Beig–Schmidt gauge, allowing us to address the dynamical evolution taking place between $\mathscr{I}_+^-$ and $\mathscr{I}_-^+$. The Bondi data maps onto a restricted subset of the Beig–Schmidt data, yielding in particular a vanishing leading magnetic Weyl tensor $B_{ab}$ at $i^0$. This justifies to some extend the assumption made by Prabhu and Shehzad who provided a different derivation of charge matching between $i^0$ and $\mathscr{I}$ [24]. We also confirmed that the electric and magnetic potentials $\sigma$ and $k_{ab}$ satisfy specific parity properties on the hyperboloid $\mathcal{H}$ (similar parity properties naturally arise in the Hamiltonian framework [40–42]).

Several comments are in order along with a cursory overview of future directions.

**General $u$-behavior at $\mathscr{I}$.** We made specific assumptions regarding the falloff rate of the matter stress tensor (2.5) and other gravitational quantities (2.4) in the limit to $\mathscr{I}_-^+$. While it has been recognised long ago that $m^0$ should not be restricted to be spherically symmetric – as opposed to what characterise CK spacetimes [17] – in order to not trivialize BMS charges [23], the conditions on the asymptotic behaviour of the shear/news tensor are much subtler.

We start from the assumption that the shear is expanded in a Taylor series in $u^{-1}$. The picture we present is in line with the minimal requirements for recovering the subleading soft graviton theorem. The condition $N_{AB} = o(u^{-\gamma})$ is usually taken in order to guarantee that the associated BMS fluxes are finite on all of $\mathscr{I}$. Above $\gamma = 1$ for the leading soft theorem [4], $\gamma = 2$ for the subleading soft theorem [49], and $\gamma = 3$ for the sub-subleading soft theorem [60]. However, it is important to stress that our work does not exclude other possibilities that have been given both in the context of mathematical general relativity and soft theorems, as exemplified below.

For example, Christodoulou–Klainerman proof of the non-linear stability of Minkowski space implies $N_{AB} = O(u^{-3/2})$ [17], while both different stability proofs [61,62] and the work of Prabhu and Shezad result in less stringent falloff behaviours [23,24]. We can compare such various proposal with our working hypotheses by noticing that the crucial argument given at the end of Section 2 still sets to zero $C_{AB}^\alpha$ in $C_{AB} = C_{AB}^0 + u^{-\alpha}C_{AB}^\alpha + \ldots$ for any $\alpha \in (0,1)$. Furthermore, it is clear from our map that Bondi data with non-integer $\alpha$ are mapped to

phase spaces at spacelike infinity that differ from the standard Beig-Schmidt phase space because of necessarily non-integer powers of $\rho$ and $\tau$ needed in the map.

Similarly, the vanishing of the $O(u^{-1})$ term in the shear $C_{AB}$ can be contrasted with the appearance of an analogous term in linearised gravity in conjunction with logarithmic corrections to the subleading soft graviton theorem [63,64]. According to (2.3b), this term would yield a $O(\ln u)$ term in the angular momentum aspect, which is excluded in our non-polyhomogeneous falloff conditions (2.4). It further appears that such logarithmic tails would yield divergent Lorentz charges (6.4) in the limit to $\mathscr{I}^+_-$.

Connected to this, but not only tied to it, the assumption that the large $u$ expansion is polynomial both for $C_{AB}$ and the other metric functions that we consider, as well as for all $r$-subleading terms of the metric, does not hold in general [65,66]. While polyhomogeneous asymptotic expansions in $r$ are believed to be a generic feature of physically relevant asymptotically flat systems (of which CK spacetimes are an example) [67–69], although contrasting results exist (see [70] and references therein), the polyhomogeneous behaviour in $u$ is less understood. The potential interconnection of the two sources of polyhomogeneity has been briefly recognised in [65]. It is clear from the present work that more general Beig–Schmidt data than those we have considered might result in such configurations at null infinity.

Further investigation of all these points are clearly desired. As a guiding principle, one could hope to deal with the issue of flux divergences in the $u$-integrals not by imposing ad hoc conditions, however well motivated, but by developing a suitable renormalization scheme. This is currently missing.

Independently of the issue of $u$-renormalization, a typical question within mathematical general relativity is that concerning the existence of physically relevant configurations that satisfy given conditions on the asymptotic structure or, somewhat equivalently, the existence of well-defined Cauchy data that evolve to a given asymptotic structure. The works on the non-linear stability of Minkowski spacetime are pivotal examples. Recently, Mohamed and Valiente Kroon studied the interplay between initial data sets of spin-1 and spin-2 fields and matching of the corresponding asymptotic charges across spatial infinity [19]. In some sense, the philosophy of this latter work is reverse to ours, as they assess which subset of asymptotic initial data gives rise to finite charges at the corners of $\mathscr{I}$, while we prescribe conditions at $\mathscr{I}$ such that charges are finite and reconstruct the corresponding data at spatial infinity, which turns out to have restricted parity properties.

**Sub-subleading antipodal matching.** The antipodal matching relations provided in (5.1) do not suffice to derive the sub-subleading soft graviton theorem [10, 11, 60, 71]. For

this one needs an extra antipodal matching condition on one of the subleading Bondi fields sitting at order $O(r^{-1})$ in the angular components of the metric [60]. An obvious continuation of our work would be to work out the dictionary between Bondi and Beig–Schmidt data to lower orders in $r$ and $u$ such as to access the relevant field. It seems very likely that the extra antipodal matching relation would only hold under the stronger assumption $N_{AB} = o(u^{-3})$ used in [60].

**Extensions of BMS and corresponding phase space.** We restricted our attention to the standard Bondi phase space in which the metric on the sphere $\mathbb{S}^2$ is the smooth unit round sphere metric. This constraint should be relaxed in order to allow for extensions of the BMS group at $\mathscr{I}$ that also include *superrotations* [6–11]. However it is not known whether spatial infinity $i^0$ also admits such extensions of the BMS group, and if it exists, the corresponding extended phase space in Beig–Schmidt gauge is yet to be uncovered. This is the reason we did not discuss the matching of superrotation charges between $\mathscr{I}$ and $i^0$ in section 6 ; none has been defined at $i^0$ as of yet. Note however that the antipodal matching relations (5.1) do imply superrotation charge conservation across $i^0$ even without an explicit matching to conserved charges at $i^0$. We hope to report on extended BMS symmetries at spatial infinity in the future.

# Acknowledgments

We thank Geoffrey Compère and Romain Ruzziconi for their reading of the manuscript and useful comments. We thank Piotr Chrusciel, Marc Henneaux, Prahar Mitra, Kostas Skenderis and Juan Valiente Kroon for valuable discussions. KN thanks Jakob Salzer for past collaboration on related topics. The work of KN was supported by the STFC grants ST/P000258/1 and ST/T000759/1. The work of EP was supported by the Royal Society Research Grants RGF/EA/181054 and RF/ERE/210267. FC is funded by the EPSRC Doctoral Prize Award EP/T517859/1.

# A    Late-time expansions

We work out the behavior of the Beig–Schmidt fields $\sigma$, $k_{ab}$ and $j_{ab}$ in the asymptotic past and future of the de Sitter hyperboloid $\mathcal{H}$, corresponding to the limits $\tau \to \pm\infty$ in the global coordinate system $(\tau, x^A)$.

**Electric potential.** The equation of motion (3.8) of $\sigma$ takes the form

$$\left(-\partial_\tau^2 - 2\tanh\tau\,\partial_\tau + \cosh^{-2}\tau\,\nabla^2 + 3\right)\sigma = 0\,. \tag{A.1}$$

The corresponding asymptotic solution is found to be

$$\sigma(\tau, x) = e^\tau\,\sigma^{(-1)} + e^{-\tau}\sigma^{(1)} + e^{-3\tau}\tau\,\tilde\sigma + e^{-3\tau}\,\sigma^{(3)} + \ldots\,, \tag{A.2}$$

where $\sigma^{(-1)}$ and $\sigma^{(3)}$ are free functions that specify a solution to the quadratic differential equation (A.1), while all other functions can be fully determined,

$$\sigma^{(1)} = -\left(\nabla^2 + 1\right)\sigma^{(-1)}\,, \qquad \tilde\sigma = \nabla^2\left(\nabla^2 + 2\right)\sigma^{(-1)}\,, \qquad \ldots \tag{A.3}$$

**Mathematical interlude.** In order to streamline the analysis of $k_{ab}$ and $j_{ab}$ to come momentarily, we consider the generic inhomogeneous differential equation for a symmetric tensor $t_{ab}$,

$$\left(D^2 - \alpha\right)t_{ab} = S_{ab}\,, \qquad \alpha \in \mathbb{R}\,, \tag{A.4}$$

in the case where not only the source $S_{ab}$ but also the trace $t_a^a$ and divergence $D^a t_{ab}$ are non-dynamical and pre-determined quantities. This situation will apply to both $k_{ab}$ and $j_{ab}$. The trace and divergence of $t_{ab}$ can be written

$$t_a^a = -t_{\tau\tau} + h^{AB}t_{AB}\,,, \tag{A.5a}$$

$$D^a t_{a\tau} = -\left(\partial_\tau + 2\tanh\tau\right)t_{\tau\tau} - \tanh\tau\,h^{AB}t_{AB} + h^{AB}\nabla_A t_{B\tau}\,, \tag{A.5b}$$

$$D^b t_{bA} = -\left(\partial_\tau + 2\tanh\tau\right)t_{A\tau} + h^{BC}\nabla_B t_{CA}\,, \tag{A.5c}$$

or equivalently

$$h^{AB}t_{AB} = t_{\tau\tau} + t_a^a\,, \tag{A.6a}$$

$$h^{AB}\nabla_A t_{B\tau} = \left(\partial_\tau + 3\tanh\tau\right)t_{\tau\tau} + D^b t_{b\tau} + \tanh\tau\,t_a^a\,, \tag{A.6b}$$

$$h^{BC}\nabla_B t_{CA} = \left(\partial_\tau + 2\tanh\tau\right)t_{A\tau} + D^b t_{bA}\,. \tag{A.6c}$$

These equations allow to eliminate the combinations on the left-hand side in terms of $t_{\tau a}$ and the non-dynamical quantities $t_a^a$ and $D^a t_{ab}$. Introducing the differential operators

$$D_1 \equiv -\partial_\tau^2 - 6\tanh\tau\,\partial_\tau - 6\tanh^2\tau + \cosh^{-2}\tau\,\nabla^2\,, \tag{A.7a}$$

$$D_2 \equiv -\partial_\tau^2 - 2\tanh\tau\,\partial_\tau + 2\tanh^2\tau + \cosh^{-2}\tau\left(1 + \nabla^2\right)\,, \tag{A.7b}$$

$$D_3 \equiv -\partial_\tau^2 + 2\tanh\tau\,\partial_\tau + \cosh^{-2}\tau\,\nabla^2 + 2\,, \tag{A.7c}$$

and making use of (A.6), the equations of motion (A.4) become

$$(D_1 - \alpha)\, t_{\tau\tau} = S_{\tau\tau} + 4 \tanh \tau \, D^b t_{b\tau} + 2 \tanh^2 \tau \, t_a^a \,, \tag{A.8a}$$

$$(D_2 - \alpha)\, t_{\tau A} = S_{\tau A} + 2 \tanh \tau \, \partial_A t_{\tau\tau} + 2 \tanh \tau \, D^b t_{bA} \,, \tag{A.8b}$$

$$(D_3 - \alpha)\, t_{AB} = S_{AB} + 2 \tanh \tau \left( \nabla_A t_{B\tau} + \nabla_B t_{A\tau} \right) . \tag{A.8c}$$

They can be solved each one in turn. Indeed (A.8a) is a simple inhomogeneous ordinary differential equation governing the dynamics of the component $t_{\tau\tau}$, with all quantities on the right-hand side being pre-determined functions. Once the solution for $t_{\tau\tau}$ has been found, one can go on and solve (A.8b) in order to determine $t_{\tau A}$, and then similarly solve (A.8c) in order to determine $t_{AB}$ .

**Magnetic potential.** The magnetic potential $k_{ab}$ satisfies the equation of motion (A.4) with $\alpha = 3$ and vanishing source term, trace and divergence. Solving (A.8) asymptotically we find

$$k_{\tau\tau} = e^{-3\tau} \tau \, \tilde{k}_{\tau\tau} + e^{-3\tau} \, k_{\tau\tau}^{(3)} + \dots \,, \tag{A.9a}$$

$$k_{\tau A} = e^{-\tau} \tau \, \tilde{k}_{\tau A} + e^{-\tau} \, k_{\tau A}^{(1)} + \dots \,, \tag{A.9b}$$

$$k_{AB} = e^{\tau} \tau \, \tilde{k}_{AB} + e^{\tau} \, k_{AB}^{(-1)} + \dots \,, \tag{A.9c}$$

where all functions appearing are free data which serve to specify a particular solution, while all subleading terms can be determined order by order. Note that these asymptotics are those of the homogeneous solutions, while the dependencies on the terms on the right-hand side of (A.8) appear at subleading order in these expansions.

**Subleading field.** The subleading field $j_{ab}$ satisfies the equation of motion (A.4) with $\alpha = 2$ and pre-determined but nonzero source, trace and divergence given in (3.10)-(3.11). Generic solutions involve superpositions of homogeneous solutions and particular solutions depending on the pre-determined quantities appearing on the right-hand side of (A.8). The asymptotic behavior of the homogeneous solutions is straightforward to determine,

$$j_{\tau\tau} = e^{-2\tau} j_{\tau\tau}^{(2)} + e^{-4\tau} j_{\tau\tau}^{(4)} + \dots \,, \tag{A.10a}$$

$$j_{\tau A} = j_{\tau A}^{(0)} + e^{-2\tau} j_{\tau A}^{(2)} + \dots \,, \tag{A.10b}$$

$$j_{AB} = e^{2\tau} j_{AB}^{(-2)} + j_{AB}^{(0)} + \dots . \tag{A.10c}$$

On the other hand, the asymptotic behavior of the particular solutions strongly depends on the asymptotic behavior of the first order fields $\sigma$ and $k_{ab}$. Using the data determined from

the Bondi phase space and given in section 4, we explicitly evaluate the right-hand side of (A.8),

$$S_{\tau\tau} + 4\tanh\tau\, D^b j_{b\tau} + 2\tanh^2\tau\, j_a^a = O(e^{-6\tau}), \tag{A.11a}$$

$$S_{\tau A} + 2\tanh\tau\, \partial_A j_{\tau\tau} + 2\tanh\tau\, D^b j_{bA} = O(e^{-4\tau}), \tag{A.11b}$$

$$S_{AB} + 2\tanh\tau\, (\nabla_A j_{B\tau} + \nabla_B j_{A\tau}) = O(e^{-2\tau}). \tag{A.11c}$$

This means that the particular solutions of $j_{ab}$ are subleading in $\tau$ compared to the homogeneous solutions. Thus (3.16) indeed describes the asymptotic behavior of a generic solution which can be mapped to the Bondi phase space. Note that some of the subleading Bondi data which we have not explicitly considered in this work would in principle contribute to (A.11a) at order $O(e^{-4\tau})$, but for consistency such terms must cancel out.

# B  Details of the map between Bondi and Beig–Schmidt data

In this appendix we present the details of the doubly-asymptotic coordinate transformation used to map the Bondi data of section 2 onto the Beig-Schmidt data described in section 3. The resulting map is summarised in section 4.

As explained in the main text, the idea is to map the second order Bondi metric (2.1) to the Beig-Schmidt gauge, to second order in $1/\rho$ and to leading order in $\tau$. We consider an appropriate ansatz for the transformation between Bondi coordinates $(u, r, x^A)$ and Beig–Schmidt coordinates $(\rho, \tau, y^A)$,

$$u = -\rho\, e^{-\tau} + \alpha(\tau, y^A) + \frac{A(\tau, y^A)}{\rho} + ..., \tag{B.1a}$$

$$r = \rho\cosh\tau + \beta(\tau, y^A) + \frac{B(\tau, y^A)}{\rho} + ..., \tag{B.1b}$$

$$x^A = y^A + \frac{p^A(\tau, y^A)}{\rho} + \frac{q^A(\tau, y^A)}{\rho^2} + ..., \tag{B.1c}$$

where $\alpha, A, \kappa, B, p^A$ and $q^A$ are arbitrary functions on the hyperboloid that are to be determined order by order in $\rho$ by enforcing the Beig-Schmidt gauge. At leading order this transformation coincides with (4.1), i.e. it relates Minkowski space written in retarded coordinates and in hyperbolic foliation. Note also that although the sphere coordinates $x^A$ and $y^A$ differ by terms which vanish in the limit $\rho \to \infty$, they can be swapped at will once the

mapping of fields at a given order in $\rho$ has been determined. This allows us to write down the dictionary in a way that is manifestly covariant on the (celestial) sphere $\mathbb{S}^2$.

The Beig–Schmidt gauge conditions to be imposed at the relevant order in $\rho$ consist in

$$g_{\rho\rho} = 1 + \frac{2\sigma}{\rho} + \frac{\sigma^2}{\rho^2} + o(\rho^{-2}), \qquad g_{\rho a} = o(\rho^{-1}). \tag{B.2}$$

At first order in $\rho$, this relates $\sigma$ to the Bondi mass aspect according to

$$\sigma(\tau, y) = 2m^0 e^{-3\tau} + \dots, \tag{B.3}$$

while to leading order in $\tau$ one finds for the coordinate transformation

$$\alpha(\tau, y) = 8m^0 \left(\tau - \frac{1}{3}\right) e^{-4\tau} + \dots, \tag{B.4a}$$

$$\beta(\tau, y) = 8m^0 \left(\tau - \frac{1}{3}\right) e^{-2\tau} + \dots, \tag{B.4b}$$

$$p^A(\tau, y) = -2\nabla_B C_0^{AB} e^{-3\tau} + \dots. \tag{B.4c}$$

This entails

$$k_{\tau\tau} = \frac{8}{3} m^0 (24\tau - 17) e^{-3\tau} + \dots, \tag{B.5a}$$

$$k_{\tau A} = 2\nabla^B C_{AB}^0 e^{-\tau} + \dots, \tag{B.5b}$$

$$k_{AB} = \frac{1}{2} C_{AB}^0 e^{\tau} + \dots. \tag{B.5c}$$

In a similar way, the functions in the second order transformation are found to be

$$A(\tau, y) = \left(\nabla_E C_{AB}^0 \nabla^A C_0^{EB} - \nabla_E C_{AB}^0 \nabla^E C_0^{AB} + 4\phi\right) e^{-5\tau} + \dots, \tag{B.6a}$$

$$B(\tau, y) = -\frac{1}{8} C_{AB}^0 C_0^{AB} e^{-\tau} + \dots, \tag{B.6b}$$

$$q^A(\tau, y) = 2\gamma^{AB} \left(-\frac{4}{3} N_B^0 + C_0^{EF} \nabla_B C_{EF}^0 - C_0^{EF} \nabla_E C_{BF}^0\right) e^{-4\tau} + \dots. \tag{B.6c}$$

From the resulting Beig-Schmidt metric we can read off the leading data of $j_{ab}$,

$$j_{\tau\tau} = -4 \left(\nabla_E C_{AB}^0 \nabla^A C_0^{EB} - \nabla_E C_{AB}^0 \nabla^E C_0^{AB} - 16\phi^0\right) e^{-4\tau} + \dots, \tag{B.7a}$$

$$j_{\tau A} = \left(4N_A^0 + C_{AB}^0 \nabla_C C_0^{BC}\right) e^{-2\tau} + \dots, \tag{B.7b}$$

$$j_{AB} = \frac{1}{8} C_{EF}^0 C_0^{EF} \gamma_{AB} + \left(-\frac{4}{3} \nabla_A N_B^0 + 8\left(\tau - \frac{1}{3}\right) m^0 C_{AB}^0 + U_{AB}^0\right) e^{-2\tau} + \dots, \tag{B.7c}$$

where $U_{AB}$ is defined as the following tracefree tensor on the 2-sphere,

$$
\begin{aligned}
U_{AB} = &-(\nabla^E \nabla^F C_{EF}) C_{AB} - \nabla_E C^{EF} \nabla_F C_{AB} + \nabla_E C_{FA} \nabla_B C^{EF} \\
&+ C^{EF} \nabla_E \nabla_F C_{AB} - C^{EF} \nabla_E \nabla_A C_{FB} - \text{trace} ,
\end{aligned}
\tag{B.8}
$$

and $\phi^0$ is the order $O(u^0)$ term in

$$
\phi = \frac{1}{2} \left( R_{(2)} - \nabla^A g_{(1)uA} + \nabla^2 g_{(2)ur} \right) \equiv u\,\phi^{-1} + \phi^0 + o(u^0) .
\tag{B.9}
$$

The coefficients $g_{(1)uA}$ and $g_{(2)ur}$ are the orders $O(r^{-1})$ and $O(r^{-2})$ of the corresponding metric components (2.2c) and (2.2b), and $R_{(2)} = \gamma^{AB} R_{(2)AB} - C^{AB} R_{(1)AB} + \bar{g}_{(0)}^{AB} R_{(0)AB}$ is the order $O(r^{-2})$ in the asymptotic expansion of the Ricci scalar associated with $g_{AB}$ ($\bar{g}_{(0)}^{AB}$ the inverse of the order $O(r^0)$ in $g_{AB}$). One can show that $R_{(2)}$ only depends on $\gamma_{AB}$ and $C_{AB}$ since $g_{(0)AB}$ does not have a $C_{AB}$-independent trace-free part.

# C   Vanishing of the leading magnetic Weyl tensor

In this appendix we give the form of $k_{ab}$ in case where the leading magnetic Weyl tensor $B_{ab}$ vanishes, and we work out the corresponding large-$\tau$ expansion. This allows us to demonstrate that the Beig–Schmidt data (4.5) obtained by explicit coordinate transformation from Bondi gauge is that associated with a vanishing $B_{ab}$. We also confirm the identification between the BMS supertranslation Goldstone mode and the Spi-supertranslation Goldstone mode that was previously made in [50].

The vanishing of the leading magnetic part of the Weyl tensor $B_{ab}$ defined in (3.5) is equivalent to the condition

$$
D_{[a} k_{b]c} = 0 .
\tag{C.1}
$$

On the three-dimensional hyperboloid $\mathcal{H}$ a symmetric traceless tensor satisfying the above condition can be written in terms of a scalar potential $\Phi$ [25, 72, 73],

$$
k_{ab} = - \left( D_a D_b + h_{ab} \right) \Phi , \qquad \left( D^2 + 3 \right) \Phi = 0 .
\tag{C.2}
$$

The scalar field $\Phi$ is the Goldstone mode of *Spi-supertranslations* [50]. Proceeding exactly as we did with the electric potential $\sigma$ in appendix A, we find its large-$\tau$ expansion to be

$$
\Phi(\tau, x) = e^\tau \Phi^{(-1)} + e^{-\tau} \Phi^{(1)} + e^{-3\tau} \tau\, \tilde{\Phi} + e^{-3\tau} \Phi^{(3)} + ... ,
\tag{C.3}
$$

with

$$
\Phi^{(1)} = - \left( \nabla^2 + 1 \right) \Phi^{(-1)} , \qquad \tilde{\Phi} = \nabla^2 \left( \nabla^2 + 2 \right) \Phi^{(-1)} , \qquad ...
\tag{C.4}
$$

Now we express (C.2) in global coordinates $(\tau, x^A)$,

$$k_{\tau\tau} = \left(-\partial_\tau^2 + 1\right)\Phi\,, \tag{C.5a}$$

$$k_{\tau A} = \left(-\partial_\tau + \tanh\tau\right)\partial_A\Phi\,, \tag{C.5b}$$

$$k_{AB} = \left(\gamma_{AB}\cosh^2\tau\left(\tanh\tau\,\partial_\tau - 1\right) - \nabla_A\nabla_B\right)\Phi\,, \tag{C.5c}$$

and plug in the asymptotic expansion (C.3) of the scalar potential $\Phi$. Doing so we find that the leading order terms for the various components of $k_{ab}$ are

$$\tilde{k}_{\tau\tau} = -8\,\tilde{\Phi} = 8\,\nabla^A\nabla^B C^\Phi_{AB}\,, \tag{C.6a}$$

$$k^{(3)}_{\tau\tau} = -8\,\Phi^{(3)}\,, \tag{C.6b}$$

$$\tilde{k}_{\tau A} = 0\,, \tag{C.6c}$$

$$k^{(1)}_{\tau A} = -2\,\partial_A(\Phi^{(-1)} - \Phi^{(1)}) = 2\,\nabla^B C^\Phi_{AB}\,, \tag{C.6d}$$

$$\tilde{k}_{AB} = 0\,, \tag{C.6e}$$

$$k^{(-1)}_{AB} = -\nabla_A\nabla_B\Phi^{(-1)} - \frac{1}{2}\gamma_{AB}\left(\Phi^{(-1)} + \Phi^{(1)}\right) = \frac{1}{2}C^\Phi_{AB}\,, \tag{C.6f}$$

where we have suggestively defined

$$C^\Phi_{AB} \equiv -2\,\nabla_A\nabla_B\Phi^{(-1)} + \gamma_{AB}\nabla^2\Phi^{(-1)}\,. \tag{C.7}$$

We observe that (C.6) perfectly agree with (4.5) provided that we make the identification

$$\Phi^{(-1)} = C\,, \tag{C.8}$$

which in particular also implies $C^\Phi_{AB} = C^0_{AB}$. Building on the work of Troessaert who pointed out that BMS supertranslations are isomorphic to Spi-supertranslations [39], (C.8) had been recently argued to hold since both members transform in the same way [50]. Since we now have the map (4.5) between the large-$\tau$ behavior of $k_{ab}$ and the supertranslation mode $C$, we are able to confirm the identification (C.8) without relying on their transformation properties, while at the same time proving a consistency check of our findings. Finally, note that it is known that the subleading mode $\Phi^{(3)}$ can always be removed by a pure gauge transformation [39], and we can therefore set it to zero without loss of generality.

The data $\tilde{k}_{\tau\tau}, k^{(3)}_{\tau\tau}, \tilde{k}_{\tau A}, k^{(1)}_{\tau A}, \tilde{k}_{AB}, k^{(-1)}_{AB}$ fully specifies a solution for $k_{ab}$. Since the data (4.5) obtained in the main text can be written in the form (C.6), we conclude that the full solution of $k_{ab}$ also admits the form (C.2). This further implies the vanishing of the leading magnetic Weyl tensor *whenever the solution can be mapped onto the Bondi phase space*,

$$B_{ab} = 0\,. \tag{C.9}$$

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
