# Peer review of "Charge and Antipodal Matching across Spatial Infinity"

_SciPost Physics_

## Round 1 · Referee Report · Anonymous (Referee 1) · 2022-6-7

Strengths

  1. Clarity of presentation.
  2. Organization of presentation.

Report

The map between spatial infinity quantities and null infinity quantities is a much-discussed subject in recent years. There are many approaches different authors have taken. It is in this context that the paper under review works out a map between Bondi and Beig-Schmidt gauges by performing an asymptotic coordinate transformation. The authors take an explicit coordinate-based approach.

The work is inspired by celestial holography literature. The authors assume that the scattering data at null infinity admits a polynomial expansion in negative powers of the radial and retarded time coordinates. Under these assumptions, they find how the scattering data maps onto a Beig--Schmidt gauge.

The paper builds upon earlier literature on the subject, especially on the work by Troessaert et al and Prabhu et al. It provides a complete map of asymptotic data and charges, taking care of non-linearities under the above assumptions.

From their analysis, it also follows that the various proposals for the BMS charges in Bondi gauge all match with the conserved charges at spatial infinity. The terms by which various proposals differ all vanish in the limit to spatial infinity under the assumptions of the paper.

The key technical results of the paper are:

  1. Bondi data maps onto a subset of the studied Beig--Schmidt data, in such a way that the large $\tau$ behavior of the particular solution is sub-leading compared to that of the homogeneous solutions. This result provides support to various earlier claims in the literature. The authors view this separation between the large-$\tau$ asymptotics of the homogeneous and particular solutions as a feature of the Bondi gauge. This observation plays a crucial to the derivation of the corresponding antipodal matching condition for the angular momentum aspect studied in a later section.

  2. Another result of the paper is that in going from Bondi to Beig--Schmidt gauge under the working assumptions of the authors, the logarithmic term is not generated. Moreover, the map is such that the leading order magnetic part of the Weyl tensor vanishes, and the potential $k_{ab}$ of the leading order magnetic part of the Weyl tensor takes the form of a double derivative on a scalar. The authors argue that $k_{ab}$ is thereby fully determined by the supertranslation mode $C$.

Overall, the paper is nicely written. It contains several interesting calculations. It provides a synergetic link between different papers on the subject.

Journal's acceptance criteria are met. I recommend the publication of the paper in its present form.

At one place on page 23: extend -> extent.

Requested changes

OPTIONAL:

The authors assume that the scattering data at null infinity admit a polynomial expansion in negative powers of the radial and retarded time coordinates. The authors are familiar with the literature that argues that this is perhaps not ideal. They do write two paragraphs in the conclusions about u^{-1} and log u tails in the context of the subleading soft graviton theorem. Perhaps they are also familiar with the papers by Kehrberger and an earlier argument by Christodoulou that smooth null infinity fails to accurately capture the structure of gravitational radiation emitted by infalling masses coming from past timelike infinity.

I suggest they add a small cautionary paragraph to this effect in section 2 so that the reader gets the correct impression of the scope of the analysis at the beginning itself.

  • validity: high
  • significance: high
  • originality: good
  • clarity: top
  • formatting: perfect
  • grammar: perfect

Author:  Kevin Nguyen  on 2022-07-12  [id 2654]

(in reply to Report 1 on 2022-06-07)

We thank the referee for the very positive comments on our paper and the useful suggestions. Following the referee's advice, we have added a footnote on page 5 drawing the reader’s attention to the potential limitations of non-polyhomogeneous expansions in more general contexts, including new references to Christodoulou and Kehrberger’s works. We also explicitly refer the reader to section 7 where these points are also addressed.

---

## Round 1 · Referee Report · Anonymous (Referee 2) · 2022-6-9

Strengths

  1. Clarity of the presentation.

  2. Simplicity of the approach.

  3. Concrete relation between spatial infinity and null infinity through an explicit diffeomorphism relating Beig-Schmidt and Bondi gauges.

Weaknesses

  1. Strong assumptions on the falloffs in u.

Report

In this article, the authors investigate the antipodal matching conditions between $\mathscr{I}^+_-$ and $\mathscr{I}^-_+$ by working at spatial infinity in Beig-Schmidt coordinates. In this coordinate system, spatial infinity corresponds to a three-dimensional de Sitter hyperboloid along which one can consider a time evolution. Requiring some falloff conditions in the infinite future and past of the hyperboloid that are needed for smoothness of null infinity, the authors select a branch of the solution space that realizes the antipodal matching. The solution space at spatial infinity is then related to the one at null infinity by constructing an explicit diffeomorphism between Beig-Schmidt and Bondi gauges. The BMS charges at $\mathscr{I}^+_-$ and $\mathscr{I}^-_+$ are matched with the conserved BMS charges at spatial infinity.

The relation between soft theorems and BMS Ward identities requires the antipodal matching between $\mathscr{I}^+_-$ and $\mathscr{I}^-_+$ as an assumption. Trying to demonstrate this antipodal matching by an investigation of the gravitational phase space at spatial infinity is definitely worth pursuing. Though some results along these lines have already been obtained in the literature (see e.g. references [23], [24], [39], [40]), the framework used in this paper clarifies some aspects of this derivation, especially the matching of the BMS charges between spatial infinity and null infinity. Therefore this article meets SciPost acceptance criteria and I am happy to recommend it for publication in this journal.

Requested changes

OPTIONAL:

The authors might find useful to consider the following comments:

  1. The electricity condition (2.6), which is a common assumption in the celestial holography literature, implies that the solution space does not include the Taub-NUT metric. However, due to the recent interest for dual gravitational charges and their implications on soft graviton theorems (see e.g. 1812.01641, 1907.00990, 1908.01164), it would be worth relaxing this condition. Notice that the boundary conditions at spatial infinity considered in 1805.11288 accommodate the Taub-NUT solution (by contrast with those originally proposed in 1801.0371). In this sense, the boundary conditions considered here are more restrictive than those of 1805.11288.

  2. In Beig-Schmidt gauge, $\mathscr{I}^+_-$ and $\mathscr{I}^-_+$ are reached by taking a double limit on the coordinates, together with some hierarchy in the order of the limits. This makes the interpretation of the matching between null infinity and spatial infinity a bit hard to visualize. Indeed, it is not clear if $\mathscr{I}^+_-$ and $\mathscr{I}^-_+$ correspond to the infinite future and past of the de Sitter hyperboloid at infinity, which is what one might have naively expected. As already noted in 1704.06223, this would have been the case if one had considered the Friedrich coordinates (H. Friedrich, Einstein equations and conformal structure: Existence of anti-de Sitter-type space-times, Journal of Geometry and Physics 17 (1995) or H. Friedrich, Gravitational fields near space-like and null infinity, Journal of Geometry and Physics 24 (1998)). It might be interesting to comment on the relation between Beig-Schmidt and Friedrich coordinates (see e.g. 2103.02389).

  • validity: high
  • significance: good
  • originality: good
  • clarity: high
  • formatting: excellent
  • grammar: excellent

Author:  Kevin Nguyen  on 2022-07-12  [id 2655]

(in reply to Report 2 on 2022-06-09)

We thank the referee for the positive comments on our paper and the useful suggestions. Following the referee's advice, we have added a sentence on page 13 drawing the reader’s attention to the fact that our set-up excludes Taub-NUT solutions, as is usual when one imposes the electricity condition on the shear tensor towards i0. We also added a footnote on page 12 commenting on the relation with Friedrich coordinates.

---

## Round 1 · Referee Report · Anonymous (Referee 3) · 2022-6-21

Report

In this paper the author consider the matching of charges associated to the BMS symemtries between null infinity and spatial infinity. They use the Bondi coordinates for asymptotics near null infinity and Beig-Schimdt coordinates for asymptotics near spatial infinity. The result of the charge matching is then claimed to follow from converting from the Bondi coordinate expansion to the Beig-Schmidt coordiante expansion of the metric.

This approach entirely ignores the key issue involved in proving these matching conditions, which is that the unphysical spacetime (after conformal completion of the physical spacetime) is not smooth at spatial infinity. In fact, the differential structure at spatial infinity is very weak, as was clarified by Geroch (for initial data sets) and by Ashtekar-Hansen inn spacetime terms. If one considers asymptotics near spatial infinity in spatial directions, the Beig-Schimdt asymptotic expansion is equivalent to the Ashtekar-Hansen structure. However the differential structure and the behaviour of the metric in null directions at spatial infinity is unknown in general.

The coordinate transformations from Bondi coordinates to Beig-Schmidt coordinate used in the paper implicitly assumes that the spacetime is C^{>1} at spatial infinity in both null and spatial directions. Further, converting between the asymptotic expansions of the metric in each coordinate system as done in the paper, implicitly assumes that the (conformally completed) metric is C^{>0} in both null and spatial directions. While these assumptions are valid in Kerr-Newman spacetimes, they are not known to be valid in general spacetimes in nonlinear general relativity. This is the key issue in showing the matching of the charges which is entirely ignored by the authors by simply assuming that such coordinate transformations work in general. The authors do not justify these assumptions in any way. Therefore their claim that this result generalizes/extends previous results on the matching of charges is unfounded.

Further, even the falloff conditions along null infinity imposed in the paper are very suspect. For example, even if one assumes the falloff in eq. 2.4a then the falloff in eq. 2.4b does not follow from 2.3b. A direct simple computation shows that given eq.2.4a, then N_A has a logarithmic divergence from eq.2.3b. So the falloffs in eq.2.4b do not follow from the evolution equations, as is incorrectly claimed by the authors.

To summarize, the authors make many, many implicit (and sometimes incorrect) assumptions. I would suggest that the authors reconsider what they assuming versus proving much more carefully as is warranted in this problem, instead of naively doing coordinate transformations. I do not think this paper adds anything to the existing work on this problem, and very certainly does not extend existing results as claimed by the authors. Modifying the paper to be suitable would result in an entirely new paper with much more rigorous analysis. As such I recommend that this paper be rejected.
  • validity: poor
  • significance: poor
  • originality: poor
  • clarity: good
  • formatting: good
  • grammar: perfect

Author:  Kevin Nguyen  on 2022-07-12  [id 2656]

(in reply to Report 3 on 2022-06-21)

We believe that the report misrepresents our work in two aspects, that is: a) it misleads the scope of our paper b) it incorrectly discusses some of its technical details. We are further very surprised by the unconventional tone of this report.

Concerning the first issue, the referee writes
''Therefore their claim that this result generalizes/extends previous results on the matching of charges is unfounded."

and

"very certainly does not extend existing results as claimed by the authors."

We would like to stress that there is no point in the paper where we claim that we generalize/extend previous results. We rather write "shed light on the matching of BMS charges" and we claim that our analysis "completes" previous results. We explained on page 15 in which sense we have completed previous results.

Furthermore, the referee claims that we assume that the coordinate transformation we use applies in general and, according to this, the referee argues that we miss or misunderstand an entire body of literature. Again we would like to notice that there is no place in the paper where we have claimed that our map applies to general spacetimes. It rather works for spacetimes whose asymptotic structure at null infinity is as specified in section 2. We have stressed in various parts of the introduction of the paper that we stick to that case in order to highlight how assumptions about gravitational data at null infinity constrain the phase space structure at spacelike infinity (and vice versa).

Regarding the second issue, the referee dubs as suspicious our assumptions and is concerned that our analysis is flawed by a mistake in the $u$-asymptotic expansion displayed in section 2. They correctly point out that (2.4c) generates a logarithmic term in the u-expansion of the angular momentum aspect. In fact, in addition to what the referee complains about, given (2.6) and (2.7b). Altogether they imply the conditions (2.10) on the coefficients of the expansion (2.4).

We stress that (2.4) is in line with our stated assumptions: a) we do not consider logarithmic terms in the asymptotic expansions and b) we work with conditions compatible with those usually assumed in celestial holography literature starting from hep-th/1312.2229, which is the electricity condition of $C_{AB}^0$ that follows from (2.6). As a side note, this choice is compatible with the finiteness of charges and fluxes assessed in previous works [i.e. hep-th/2108.11969 and hep-th/2106.14717]. We would also like to remark that in the discussion in section 7 we had already commented on the role of the $\log u$ terms. Thus, it is not correct to state that our assumptions are not clear or 'suspicious', as we had already taken into account all the observations made by the referee on this particular point. Nonetheless, we have slightly reformulated parts of the relevant paragraph in section 2 in order to further clarify the logic and avoid such confusions.

Lastly, the referee substantially points out that if one works in a conformal approach, our assumptions are in tension with the conditions of asymptotic flatness at spacelike and null infinity defined by Ashtekar and Hansen. We note that the Ashtekar-Hansen definition is based on the standard conformal completion of Kerr spacetimes. The referee instead remarks that Kerr-Newman spacetimes have been shown to have a completion compatible with what the referee claims to underlie our construction. The referee is also aware of the fact that there is no general proof that relevant radiative spacetimes behave in the null directions at spacelike infinity as prescribed by either the Ashtekar-Hansen conditions or the conditions we assume. Thus there is no strong evidence that one or the other framework is better suited to discuss physically relevant solutions with non-stationary epochs at future or past null infinity. Furthermore, even in stationary spacetimes, the existence of a conformal completion does not preclude the existence of other completions with different properties. For instance, this was considered as an interesting problem by Herberthson in gr-qc/9712058. The fact that there exists a conformal completion of Kerr-Newman that does not satisfy the Ashtekar-Hansen definition can be seen as a justification of this philosophy and a reminder that such definitions are not a God-given truth.

We wish to reiterate that we have never claimed to work in all generality. It would be certainly interesting to assess whether there are initial data that generates spacetimes with radiation with the asymptotics we have prescribed.

---

## Editorial Decision

resubmitted